# DensePure: Understanding Diffusion Models for Adversarial Robustness

**Chaowei Xiao**[*,1,3]   **Zhongzhu Chen** [*,2]   **Kun Jin** [*,2]   **Jiongxiao Wang** [*,1]
**Weili Nie** [3]   **Mingyan Liu** [2]   **Anima Anandkumar**[3,4]   **Bo Li**[5]   **Dawn Song**[6]
[1]ASU, [2] University of Michigan, Ann Arbor, [3] NVIDIA, [4] Caltech, [5] UIUC, [6] UC Berkeley

## Abstract

Diffusion models have been recently employed to improve certified robustness through the process of denoising. However, the theoretical understanding of why diffusion models are able to improve the certified robustness is still lacking, preventing from further improvement. In this study, we close this gap by analyzing the fundamental properties of diffusion models and establishing the conditions under which they can enhance certified robustness. This deeper understanding allows us to propose a new method DensePure, designed to improve the certified robustness of a pretrained model (i.e. classifier). Given an (adversarial) input, DensePure consists of multiple runs of denoising via the reverse process of the diffusion model (with different random seeds) to get multiple reversed samples, which are then passed through the classifier, followed by majority voting of inferred labels to make the final prediction. This design of using multiple runs of denoising is informed by our theoretical analysis of the conditional distribution of the reversed sample. Specifically, when the *data* density of a clean sample is high, its conditional density under the reverse process in a diffusion model is also high; thus sampling from the latter conditional distribution can purify the adversarial example and return the corresponding clean sample with a high probability. By using the highest density point in the conditional distribution as the reversed sample, we identify the robust region of a given instance under the diffusion model's reverse process. We show that this robust region is a union of multiple convex sets, and is potentially much larger than the robust regions identified in previous works. In practice, DensePure can approximate the label of the high density region in the conditional distribution so that it can enhance certified robustness. We conduct extensive experiments to demonstrate the effectiveness of DensePure by evaluating its certified robustness given a standard model via randomized smoothing. We show that DensePure is consistently better than existing methods on ImageNet, with 7% improvement on average. Project page:https://densepure.github.io/.

## 1 Introduction

Diffusion models have been shown to be a powerful image generation tool (Ho et al., 2020; Song et al., 2021b) owing to their iterative diffusion and denoising processes. These models have achieved state-of-the-art performance on sample quality (Dhariwal & Nichol, 2021; Vahdat et al., 2021) as well as effective mode coverage (Song et al., 2021a). A diffusion model usually consists of two processes: (i) a forward diffusion process that converts data to noise by gradually adding noise to the input, and (ii) a reverse generative process that starts from noise and generates data by denoising one step at a time (Song et al., 2021b).

Given the natural denoising property of diffusion models, *empirical* studies have leveraged them for adversarial purification (Nie et al., 2022; Wu et al., 2022; Carlini et al., 2022). For instance, Nie et al. (2022) employed diffusion models for model purification, *DiffPure*. They *empirically* show that by carefully choosing the amount of Gaussian noises added during the diffusion process, adversarial

---
[*]the first four authors contributed equally

perturbations can be removed while preserving the true label semantics. Despite the significant empirical result, there is no provable guarantee of the achieved robustness. A concurrent work (Carlini et al., 2022) instantiated the randomized smoothing approach with the diffusion model to offer a *provable guarantee* of model robustness against $L_2$-norm bounded adversarial example. However, they do not provide a theoretical understanding of why and how diffusion models contribute to such nontrivial certified robustness.

**Our Approach.** We are the first to theoretically analyze the fundamental properties of diffusion models to understand why and how diffusion models enhance certified robustness. This deeper understanding allows us to propose a new method DensePure to improve the certified robustness of any given classifier more effectively using diffusion models. An illustration of the DensePure framework is provided in Figure 1, where it consists of a pretrained diffusion model and a pretrained classifier. DensePure incorporates two steps: (i) using the reverse process of the diffusion model to obtain a sample of the posterior data distribution conditioned on the adversarial input; and (ii) repeating the reverse process multiple times with different random seeds to approximate the label of the high-density region in the conditional distribution via a simple majority vote strategy. In particular, given an adversarial input, we repeatedly feed it into the reverse process of the diffusion model to get multiple reversed examples and feed them into the classifier to calculate their labels. We then apply the *majority vote* on the set of labels to get the final predicted label.

DensePure is inspired by our theoretical analysis, where we show that the reverse process of the diffusion model provides a conditional distribution of the reversed sample given an adversarial input. Sampling from this conditional distribution can enhance the certified robustness. Specifically, we prove that when the data density of clean samples is high, it is a sufficient condition for the conditional density of the reversed samples to be also high. Therefore, in DensePure, samples from the conditional distribution can recover the ground-truth labels with a high probability.

For understanding and rigorous analysis conveniently, we use the highest density point in the conditional distribution as the deterministic reversed sample for the classifier prediction. We show that the robust region for a given sample under the diffusion model's reverse process is the union of multiple convex sets, each surrounding a region around the ground-truth label. Compared with the robust region of previous work (Cohen et al., 2019), which only focuses on only *one* region with the ground-truth label, such the union of multiple convex sets has the potential to provide a much larger robust region, resulting in higher certified robustness. Moreover, the characterization implies that the size of robust regions is affected by the relative density and the distance between data regions with the ground-truth label and those with other labels.

We conduct extensive experiments on ImageNet and CIFAR-10 datasets under different settings to evaluate the certifiable robustness of DensePure. In particular, we follow the setting from Carlini et al. (2022) and rely on randomized smoothing to certify the robustness of the adversarial perturbations bounded in the $\mathcal{L}_2$-norm. We show that DensePure achieves a new state-of-the-art *certified* robustness on the standard pretrained model without further tuning any model parameters (e.g., smooth augmentation Cohen et al. (2019)). On ImageNet, it achieves a consistently higher certified accuracy, 7% improvement on average, than the existing methods among every $\sigma$ at every radius $\epsilon$.

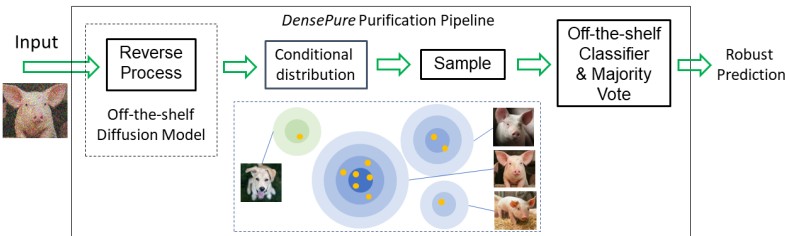

Figure 1: Pipeline of DensePure.

**Technical Contributions.** In this paper, we take the first step to understand why and how diffusion models contribute to certified robustness. We make contributions on both theoretical and empirical fronts: (1)in theory, we prove that an adversarial example can be recovered back to the original clean sample with a high probability via the reverse process of a diffusion model. (2) In theory, we characterized the robust region for each point by further taking the highest density point in the conditional

distribution generated by the reverse process as the reversed sample. We show that the robust region for a given sample under the diffusion model's reverse process has the potential to provide a larger robust region. To the best of our knowledge, this is the first work that characterizes the robust region of using the reverse process of the diffusion model for adversarial purification (3) In practice, we proposed DensePurebased on our theoretical analysis. We demonstrated DensePureis consistently better than existing methods on ImageNet, with 7% improvement on average.

## 2 PRELIMINARIES AND BACKGROUNDS

**Continuous-Time Diffusion Model.** The diffusion model has two components: the *diffusion process* followed by the *reverse process*. Given an input random variable $\mathbf{x}_0 \sim p$, the diffusion process adds isotropic Gaussian noises to the data so that the diffused random variable at time $t$ is $\mathbf{x}_t = \sqrt{\alpha_t}(\mathbf{x}_0 + \boldsymbol{\epsilon}_t)$, s.t., $\boldsymbol{\epsilon}_t \sim \mathcal{N}(\mathbf{0}, \sigma_t^2 \boldsymbol{I})$, and $\sigma_t^2 = (1 - \alpha_t)/\alpha_t$, and we denote $\mathbf{x}_t \sim p_t$. The forward diffusion process can also be defined by the stochastic differential equation

$$d\boldsymbol{x} = h(\boldsymbol{x}, t)dt + g(t)d\boldsymbol{w}, \quad \text{(SDE)}$$

where $\boldsymbol{x}_0 \sim p$, $h : \mathbb{R}^d \times \mathbb{R} \mapsto \mathbb{R}^d$ is the drift coefficient, $g : \mathbb{R} \mapsto \mathbb{R}$ is the diffusion coefficient, and $\boldsymbol{w}(t) \in \mathbb{R}^n$ is the standard Wiener process.

Under mild conditions B.1, the reverse process exists and removes the added noise by solving the reverse-time SDE (Anderson, 1982)

$$d\hat{\boldsymbol{x}} = [h(\hat{\boldsymbol{x}}, t) - g(t)^2 \nabla_{\hat{\boldsymbol{x}}} \log p_t(\hat{\boldsymbol{x}})]dt + g(t)d\overline{\boldsymbol{w}}, \quad \text{(reverse-SDE)}$$

where $dt$ is an infinitesimal reverse time step, and $\overline{\boldsymbol{w}}(t)$ is a reverse-time standard Wiener process.

In our context, we use the conventions of VP-SDE (Song et al., 2021b) where $h(\boldsymbol{x}; t) := -\frac{1}{2}\gamma(t)x$ and $g(t) := \sqrt{\gamma(t)}$ with $\gamma(t)$ positive and continuous over $[0, 1]$, such that $x(t) = \sqrt{\alpha_t}x(0) + \sqrt{1 - \alpha_t}\boldsymbol{\epsilon}$ where $\alpha_t = e^{-\int_0^t \gamma(s)ds}$ and $\boldsymbol{\epsilon} \sim \mathcal{N}(\mathbf{0}, \boldsymbol{I})$. We use $\{\mathbf{x}_t\}_{t \in [0,1]}$ and $\{\hat{\mathbf{x}}_t\}_{t \in [0,1]}$ to denote the diffusion process and the reverse process generated by SDE and reverse-SDE respectively, which follow the same distribution.

**Discrete-Time Diffusion Model (or DDPM (Ho et al., 2020)).** DDPM constructs a discrete Markov chain $\{\mathbf{x}_0, \mathbf{x}_1, \cdots, \mathbf{x}_i, \cdots, \mathbf{x}_N\}$ as the forward process for the training data $\mathbf{x}_0 \sim p$, such that $\mathbb{P}(\mathbf{x}_i|\mathbf{x}_{i-1}) = \mathcal{N}(\mathbf{x}_i; \sqrt{1 - \beta_i}\mathbf{x}_{i-1}, \beta_i I)$, where $0 < \beta_1 < \beta_2 < \cdots < \beta_N < 1$ are predefined noise scales such that $\mathbf{x}_N$ approximates the Gaussian white noise. Denote $\overline{\alpha}_i = \prod_{i=1}^{N}(1 - \beta_i)$, we have $\mathbb{P}(\mathbf{x}_i|\mathbf{x}_0) = \mathcal{N}(\mathbf{x}_i; \sqrt{\overline{\alpha}_i}\mathbf{x}_0, (1 - \overline{\alpha}_i)\boldsymbol{I})$, i.e., $\mathbf{x}_t(\mathbf{x}_0, \boldsymbol{\epsilon}) = \sqrt{\overline{\alpha}_i}\mathbf{x}_0 + (1 - \overline{\alpha}_i)\boldsymbol{\epsilon}, \boldsymbol{\epsilon} \sim \mathcal{N}(\mathbf{0}, \boldsymbol{I})$.

The reverse process of DDPM learns a reverse direction variational Markov chain $p_{\boldsymbol{\theta}}(\mathbf{x}_{i-1}|\mathbf{x}_i) = \mathcal{N}(\mathbf{x}_{i-1}; \boldsymbol{\mu}_{\boldsymbol{\theta}}(\mathbf{x}_i, i), \Sigma_{\boldsymbol{\theta}}(\mathbf{x}_i, i))$. Ho et al. (2020) defines $\boldsymbol{\epsilon}_{\boldsymbol{\theta}}$ as a function approximator to predict $\boldsymbol{\epsilon}$ from $\boldsymbol{x}_i$ such that $\boldsymbol{\mu}_{\boldsymbol{\theta}}(\mathbf{x}_i, i) = \frac{1}{\sqrt{1-\beta_i}}\left(\mathbf{x}_i - \frac{\beta_i}{\sqrt{1-\overline{\alpha}_i}}\boldsymbol{\epsilon}_{\boldsymbol{\theta}}(\mathbf{x}_i, i)\right)$. Then the reverse time samples are generated by $\hat{\mathbf{x}}_{i-1} = \frac{1}{\sqrt{1-\beta_i}}\left(\hat{\mathbf{x}}_i - \frac{\beta_i}{\sqrt{1-\overline{\alpha}_i}}\boldsymbol{\epsilon}_{\boldsymbol{\theta}^*}(\hat{\mathbf{x}}_i, i)\right) + \sqrt{\beta_i}\boldsymbol{\epsilon}, \boldsymbol{\epsilon} \sim \mathcal{N}(\mathbf{0}, I)$, and the optimal parameters $\boldsymbol{\theta}^*$ are obtained by solving $\boldsymbol{\theta}^* := \arg\min_{\boldsymbol{\theta}} \mathbb{E}_{\mathbf{x}_0, \boldsymbol{\epsilon}}\left[||\boldsymbol{\epsilon} - \boldsymbol{\epsilon}_{\boldsymbol{\theta}}(\sqrt{\overline{\alpha}_i}\mathbf{x}_0 + (1 - \overline{\alpha}_i), i)||_2^2\right]$.

**Randomized Smoothing.** Randomized smoothing is used to certify the robustness of a given classifier against $L_2$-norm based perturbation. It transfers the classifier $f$ to a smooth version $g(\boldsymbol{x}) = \arg\max_c \mathbb{P}_{\boldsymbol{\epsilon} \sim \mathcal{N}(\mathbf{0}, \sigma^2 \boldsymbol{I})}(f(\boldsymbol{x} + \boldsymbol{\epsilon}) = c)$, where $g$ is the smooth classifier and $\sigma$ is a hyperparameter of the smooth classifier $g$, which controls the trade-off between robustness and accuracy. Cohen et al. (2019) shows that $g(x)$ induces the certifiable robustness for $\boldsymbol{x}$ under the $L_2$-norm with radius $R$, where $R = \frac{\sigma}{2}\left(\Phi^{-1}(p_A) - \Phi^{-1}(p_B)\right)$; $p_A$ and $p_B$ are probability of the most probable class and "runner-up" class respectively; $\Phi$ is the inverse of the standard Gaussian CDF. The $p_A$ and $p_B$ can be estimated with arbitrarily high confidence via Monte Carlo method (Cohen et al., 2019).

## 3 THEORETICAL ANALYSIS

In this section, we theoretically analyze why and how the diffusion model can enhance the robustness of a given classifier. We will analyze directly on SDE and reverse-SDE as they generate the same

stochastic processes $\{\mathbf{x}_t\}_{t\in[0,T]}$ and the literature works establish an approximation on reverse-SDE (Song et al., 2021b; Ho et al., 2020).

We first show that given a diffusion model, solving reverse-SDE will generate a conditional distribution based on the scaled adversarial sample, which will have high density on data region with high *data* density and near to the adversarial sample in Theorem 3.1. See detailed conditions in B.1.

**Theorem 3.1.** *Under conditions B.1, solving equation reverse-SDE starting from time $t$ and sample $\boldsymbol{x}_{a,t} = \sqrt{\alpha_t}\boldsymbol{x}_a$ will generate a reversed random variable $\hat{\mathbf{x}}_0$ with density $\mathbb{P}\left(\hat{\mathbf{x}}_0 = \boldsymbol{x} | \hat{\mathbf{x}}_t = \boldsymbol{x}_{a,t}\right) \propto p(\boldsymbol{x}) \cdot \frac{1}{\sqrt{(2\pi\sigma_t^2)^n}} \exp\left(\frac{-||\boldsymbol{x}-\boldsymbol{x}_a||_2^2}{2\sigma_t^2}\right)$, where $p$ is the data distribution, $\sigma_t^2 = \frac{1-\alpha_t}{\alpha_t}$ is the variance of Gaussian noise added at time $t$ in the diffusion process.*

*Proof.* (sketch) Under conditions B.1, we know $\{\mathbf{x}_t\}_{t\in[0,1]}$ and $\{\hat{\mathbf{x}}_t\}_{t\in[0,1]}$ follow the same distribution, and then the rest proof follows Bayes' Rule. □

Please see the full proofs of this and the following theorems in Appendix B.2.

**Remark 1.** *Note that $\mathbb{P}\left(\hat{\mathbf{x}}_0 = \boldsymbol{x} | \hat{\mathbf{x}}_t = \boldsymbol{x}_{a,t}\right) > 0$ if and only if $p(\boldsymbol{x}) > 0$, thus the generated reverse sample will be on the data region where we train classifiers.*

In Theorem 3.1, the conditional density $\mathbb{P}\left(\hat{\mathbf{x}}_0 = \boldsymbol{x} | \hat{\mathbf{x}}_t = \boldsymbol{x}_{a,t}\right)$ is high if both $p(\boldsymbol{x})$ and the Gaussian term have high values, i.e., $\boldsymbol{x}$ has high *data* density and is close to the adversarial sample $\boldsymbol{x}_a$. The latter condition is reasonable since adversarial perturbations are typically bounded due to budget constraints. So the above argument implies that a reversed sample will more likely to have the ground-truth label if data region with the ground-truth label has high *data* density. For the sake of theoretical analysis and understanding, we take the point with highest conditional density $\mathbb{P}\left(\hat{\mathbf{x}}_0 = \boldsymbol{x} | \hat{\mathbf{x}}_t = \boldsymbol{x}_{a,t}\right)$ as the reversed sample, defined as $\mathcal{P}(\boldsymbol{x}_a; t) := \arg\max_{\boldsymbol{x}} \mathbb{P}\left(\hat{\mathbf{x}}_0 = \boldsymbol{x} | \hat{\mathbf{x}}_t = \boldsymbol{x}_{a,t}\right)$. $\mathcal{P}(\boldsymbol{x}_a; t)$ is a representative of the high density data region in the conditional distribution and $\mathcal{P}(\cdot; t)$ is a deterministic purification model. In the following, we characterize the robust region for data region with ground-truth label under $\mathbb{P}(\cdot; t)$. The robust region and robust radius for a general deterministic purification model given a classifier are defined below.

**Definition 3.2** (Robust Region and Robust Radius). *Given a classifier $f$ and a point $\boldsymbol{x}_0$, let $\mathcal{G}(\boldsymbol{x}_0) := \{\boldsymbol{x} : f(\boldsymbol{x}) = f(\boldsymbol{x}_0)\}$ be the data region where samples have the same label as $\boldsymbol{x}_0$. Then given a deterministic purification model $\mathcal{P}(\cdot; \psi)$ with parameter $\psi$, we define the robust region of $\mathcal{G}(\boldsymbol{x}_0)$ under $\mathcal{P}$ and $f$ as $\mathcal{D}_{\mathcal{P}}^f(\mathcal{G}(\boldsymbol{x}_0); \psi) := \{\boldsymbol{x} : f(\mathcal{P}(\boldsymbol{x}; \psi)) = f(\boldsymbol{x}_0)\}$, i.e., the set of $\boldsymbol{x}$ such that purified sample $\mathcal{P}(\boldsymbol{x}; \psi)$ has the same label as $\boldsymbol{x}_0$ under $f$. Further, we define the robust radius of $\boldsymbol{x}_0$ as $r_{\mathcal{P}}^f(\boldsymbol{x}_0; \psi) := \max\left\{r : \boldsymbol{x}_0 + ru \in \mathcal{D}_{\mathcal{P}}^f(\boldsymbol{x}_0; \psi)\ ,\ \forall ||u||_2 \leq 1\right\}$, i.e., the radius of maximum inclined ball of $\mathcal{D}_{\mathcal{P}}^f(\boldsymbol{x}_0; \psi)$ centered around $\boldsymbol{x}_0$. We will omit $\mathcal{P}$ and $f$ when it is clear from the context and write $\mathcal{D}(\mathcal{G}(\boldsymbol{x}_0); \psi)$ and $r(\boldsymbol{x}_0; \psi)$ instead.*

**Remark 2.** *In Definition 3.2, the robust region (resp. radius) is defined for each class (resp. point). When using the point with highest $\mathbb{P}\left(\hat{\mathbf{x}}_0 = \boldsymbol{x} | \hat{\mathbf{x}}_t = \boldsymbol{x}_{a,t}\right)$ as the reversed sample, $\psi := t$.*

Now given a sample $\boldsymbol{x}_0$ with ground-truth label, we are ready to characterize the robust region $\mathcal{D}(\mathcal{G}(\boldsymbol{x}_0); \psi)$ under purification model $\mathcal{P}(\cdot; t)$ and classifier $f$. Intuitively, if the adversarial sample $\boldsymbol{x}_a$ is near to $\boldsymbol{x}_0$ (in Euclidean distance), $\boldsymbol{x}_a$ keeps the same label semantics of $\boldsymbol{x}_0$ and so as the purified sample $\mathcal{P}(\boldsymbol{x}_a; t)$, which implies that $f(\mathcal{P}(\boldsymbol{x}_a; \psi)) = f(\boldsymbol{x}_0)$. However, the condition that $\boldsymbol{x}_a$ is near to $\boldsymbol{x}_0$ is sufficient but not necessary since we can still achieve $f(\mathcal{P}(\boldsymbol{x}_a; \psi)) = f(\boldsymbol{x}_0)$ if $\boldsymbol{x}_a$ is near to any sample $\tilde{\boldsymbol{x}}_0$ with $f(\mathcal{P}(\tilde{\boldsymbol{x}}_a; \psi)) = f(\boldsymbol{x}_0)$. In the following, we will show that the robust region $\mathcal{D}(\mathcal{G}(\boldsymbol{x}_0); \psi)$ is the union of the convex robust sub-regions surrounding every $\tilde{\boldsymbol{x}}_0$ with the same label as $\boldsymbol{x}_0$. The following theorem characterizes the convex robust sub-region and robust region respectively.

**Theorem 3.3.** *Under conditions B.1 and classifier $f$, let $\boldsymbol{x}_0$ be the sample with ground-truth label and $\boldsymbol{x}_a$ be the adversarial sample, then (i) the purified sample $\mathcal{P}(\boldsymbol{x}_a; t)$ will have the ground-truth label if $\boldsymbol{x}_a$ falls into the following convex set,*

$$\mathcal{D}_{sub}(\boldsymbol{x}_0; t) := \bigcap_{\left\{\boldsymbol{x}_0': f(\boldsymbol{x}_0') \neq f(\boldsymbol{x}_0)\right\}} \left\{\boldsymbol{x}_a : (\boldsymbol{x}_a - \boldsymbol{x}_0)^\top(\boldsymbol{x}_0' - \boldsymbol{x}_0) < \sigma_t^2 \log\left(\frac{p(\boldsymbol{x}_0)}{p(\boldsymbol{x}_0')}\right) + \frac{||\boldsymbol{x}_0' - \boldsymbol{x}_0||_2^2}{2}\right\},$$

*and further, (ii) the purified sample $\mathcal{P}(\boldsymbol{x}_a; t)$ will have the ground-truth label if and only if $\boldsymbol{x}_a$ falls into the following set, $\mathcal{D}(\mathcal{G}(\boldsymbol{x}_0); t) := \bigcup_{\tilde{\boldsymbol{x}}_0 : f(\tilde{\boldsymbol{x}}_0) = f(\boldsymbol{x}_0)} \mathcal{D}_{sub}(\tilde{\boldsymbol{x}}_0; t)$. In other words, $\mathcal{D}(\mathcal{G}(\boldsymbol{x}_0); t)$ is the robust region for data region $\mathcal{G}(\boldsymbol{x}_0)$ under $\mathcal{P}(\cdot; t)$ and $f$.*

*Proof.* (sketch) (i). Each convex half-space defined by the inequality corresponds to a $\boldsymbol{x}_0'$ such that $f(\boldsymbol{x}_0') \neq f(\boldsymbol{x}_0)$ where $\boldsymbol{x}_a$ within satisfies $\mathbb{P}(\hat{\mathbf{x}}_0 = \boldsymbol{x}_0 | \hat{\mathbf{x}}_t = \boldsymbol{x}_{a,t}) > \mathbb{P}(\hat{\mathbf{x}}_0 = \boldsymbol{x}_0' \mid \hat{\mathbf{x}}_t = \boldsymbol{x}_{a,t})$. This implies that $\mathcal{P}(\boldsymbol{x}_a; t) \neq \boldsymbol{x}_0'$ and $f(\mathcal{P}(\boldsymbol{x}_a; \psi)) = f(\boldsymbol{x}_0)$. The convexity is due to that the intersection of convex sets is convex. (ii). The "if" follows directly from (i). The "only if" holds because if $\boldsymbol{x}_a \notin \mathcal{D}(\mathcal{G}(\boldsymbol{x}_0); t)$, then exists $\tilde{\boldsymbol{x}}_1$ such that $f(\tilde{\boldsymbol{x}}_1) \neq f(\boldsymbol{x}_0)$ and $\mathbb{P}(\hat{\mathbf{x}}_0 = \tilde{\boldsymbol{x}}_1 | \hat{\mathbf{x}}_t = \boldsymbol{x}_{a,t}) > \mathbb{P}(\hat{\mathbf{x}}_0 = \tilde{\boldsymbol{x}}_0 | \hat{\mathbf{x}}_t = \boldsymbol{x}_{a,t}), \forall \tilde{\boldsymbol{x}}_0$ s.t. $f(\tilde{\boldsymbol{x}}_0) = f(\boldsymbol{x}_0)$, and thus $f(\mathcal{P}(\boldsymbol{x}_a; \psi)) \neq f(\boldsymbol{x}_0)$. $\qquad\square$

**Remark 3.** *Theorem 3.3 implies that when data region $\mathcal{G}(\boldsymbol{x}_0)$ has higher data density and larger distances to data regions with other labels, it tends to have larger robust region and points in data region tends to have larger radius. Since adversarial attack typically has small magnitude, with large robust region, the adversarial sample can be recovered to the clean sample with a high probability.*

In the literature, people focus more on the robust radius (lower bound) $r(\mathcal{G}(\boldsymbol{x}_0); t)$ (Cohen et al., 2019; Carlini et al., 2022), which can be obtained by finding the maximum inclined ball inside $\mathcal{D}(\mathcal{G}(\boldsymbol{x}_0); t)$ centering $\boldsymbol{x}_0$. Note that although $\mathcal{D}_{\text{sub}}(\boldsymbol{x}_0; t)$ is convex, $\mathcal{D}(\mathcal{G}(\boldsymbol{x}_0); t)$ is generally not. Therefore, finding $r(\mathcal{G}(\boldsymbol{x}_0); t)$ is a non-convex optimization problem. In particular, it can be formulated into a disjunctive optimization problem with integer indicator variables, which is typically NP-hard to solve. One alternative could be finding the maximum inclined ball in $\mathcal{D}_{\text{sub}}(\boldsymbol{x}_0; t)$, which can be formulated into a convex optimization problem whose optimal value provides a lower bound for $r(\mathcal{G}(\boldsymbol{x}_0); t)$. However, $\mathcal{D}(\mathcal{G}(\boldsymbol{x}_0); t)$ has the potential to provide much larger robustness radius because it might connect different convex robust sub-regions into one, as shown in Figure 2.

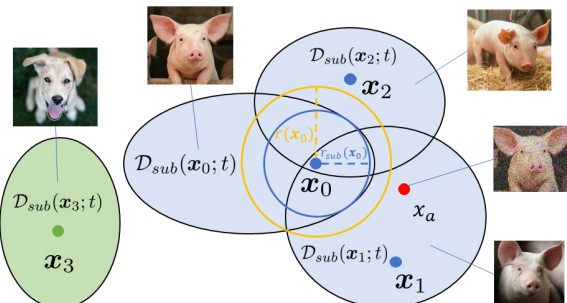

Figure 2: An illustration of the robust region $\mathcal{D}(\boldsymbol{x}_0; t) = \bigcup_{i=1}^{3} \mathcal{D}_{sub}(\boldsymbol{x}_i; t)$, where $\boldsymbol{x}_0, \boldsymbol{x}_1, \boldsymbol{x}_2$ are samples with ground-truth label and $\boldsymbol{x}_3$ is a sample with another label. $\boldsymbol{x}_a = \boldsymbol{x}_0 + \boldsymbol{\epsilon}_a$ is an adversarial sample such that $\mathcal{P}(\boldsymbol{x}_a; t) = \boldsymbol{x}_1 \neq \boldsymbol{x}_0$ and thus the classification is correct but $\boldsymbol{x}_a$ is not reversed back to $\boldsymbol{x}_0$. $r_{sub}(\boldsymbol{x}_0) < r(\boldsymbol{x}_0)$ shows our claim that the union leads to a larger robust radius.

In practice, we cannot guarantee to establish an exact reverse process like reverse-SDE but instead try to establish an approximate reverse process to mimic the exact one. As long as the approximate reverse process is close enough to the exact reverse process, they will generate close enough conditional distributions based on the adversarial sample. Then the density and locations of the data regions in two conditional distributions will not differ much and so is the robust region for each data region. We take the score-based diffusion model in Song et al. (2021b) for an example and demonstrate Theorem 3.4 to bound the KL-divergnece between conditional distributions generated by reverse-SDE and score-based diffusion model. Ho et al. (2020) showed that using variational inference to fit DDPM is equivalent to optimizing an objective resembling score-based diffusion model with a specific weighting scheme, so the results can be extended to DDPM.

**Theorem 3.4.** *Under score-based diffusion model Song et al. (2021b) and conditions B.1, we have $D_{KL}(\mathbb{P}(\hat{\mathbf{x}}_0 = \boldsymbol{x} \mid \hat{\mathbf{x}}_t = \boldsymbol{x}_{a,t}) \| \mathbb{P}(\mathbf{x}_0^\theta = \boldsymbol{x} \mid \mathbf{x}_t^\theta = \boldsymbol{x}_{a,t})) = \mathcal{J}_{\text{SM}}(\theta, t; \lambda(\cdot))$, where $\{\hat{\boldsymbol{x}}_\tau\}_{\tau \in [0,t]}$ and $\{\boldsymbol{x}_\tau^\theta\}_{\tau \in [0,t]}$ are stochastic processes generated by reverse-SDE and score-based diffusion model respectively, $\mathcal{J}_{\text{SM}}(\theta, t; \lambda(\cdot)) := \frac{1}{2} \int_0^t \mathbb{E}_{p_\tau(\mathbf{x})} \left[ \lambda(\tau) \| \nabla_{\mathbf{x}} \log p_\tau(\mathbf{x}) - \boldsymbol{s}_\theta(\mathbf{x}, \tau) \|_2^2 \right] d\tau, \boldsymbol{s}_\theta(\mathbf{x}, \tau)$ is the score function to approximate $\nabla_{\mathbf{x}} \log p_\tau(\mathbf{x})$, and $\lambda : \mathbb{R} \to \mathbb{R}$ is any weighting scheme used in the training score-based diffusion models.*

*Proof.* (sketch) Let $\boldsymbol{\mu}_t$ and $\boldsymbol{\nu}_t$ be the path measure for reverse processes $\{\hat{\mathbf{x}}_\tau\}_{\tau \in [0,t]}$ and $\{\mathbf{x}_\tau^\theta\}_{\tau \in [0,t]}$ respectively based on the $\boldsymbol{x}_{a,t}$. Under conditions B.1, $\boldsymbol{\mu}_t$ and $\boldsymbol{\nu}_t$ are uniquely defined and the KL-divergence can be computed via the Girsanov theorem Oksendal (2013). □

**Remark 4.** *Theorem 3.4 shows that if the training loss is smaller, the conditional distributions generated by reverse-SDE and score-based diffusion model are closer, and are the same if the training loss is zero. Furthermore, by the Pinsker's inequality, the total variation (a distance metric) is upper bounded by* $D_{TV}(\mathbb{P}(\hat{\mathbf{x}}_0 = \boldsymbol{x} \mid \hat{\mathbf{x}}_t = \boldsymbol{x}_{a,t}) \| \mathbb{P}(\mathbf{x}_0^\theta = \boldsymbol{x} \mid \mathbf{x}_t^\theta = \boldsymbol{x}_{a,t})) \leq \sqrt{\frac{1}{2} \mathcal{J}_{\mathrm{SM}}(\theta, t; \lambda(\cdot))}$.

## 4 DENSEPURE

Inspired by the theoretical analysis, we introduce DensePure and show how to calculate its certified robustness radius via the randomized smoothing algorithm.

**Framework.** Our framework, DensePure, consists of two components: (1) an off-the-shelf diffusion model with reverse process **rev** and (2) an off-the-shelf base classifier $f$.

The pipeline of DensePure is shown in Figure 1. Given an input $\boldsymbol{x}$, we feed it into the reverse process **rev** of the diffusion model to get the reversed sample $\mathbf{rev}(\boldsymbol{x})$ and then repeat the above process $K$ times to get $K$ reversed samples $\{\mathbf{rev}(\boldsymbol{x})_1, \cdots, \mathbf{rev}(\boldsymbol{x})_K\}$. We feed the above $K$ reversed samples into the classifier to get the corresponding prediction $\{f(\mathbf{rev}(\boldsymbol{x})_1), \cdots, f(\mathbf{rev}(\boldsymbol{x})_K)\}$ and then apply the *majority vote*, termed **MV**, on these predictions to get the final predicted label $\hat{y} = \mathbf{MV}(\{f(\mathbf{rev}(\boldsymbol{x})_1), \cdots, f(\mathbf{rev}(\boldsymbol{x})_K)\}) = \arg\max_c \sum_{i=1}^K \mathbf{1}\{f(\mathbf{rev}(\boldsymbol{x})_i) = c\}$.

**Certified Robustness of DensePure with Randomized Smoothing.**

In this paragraph, we will illustrate the algorithm to calculate certified robustness of DensePure via RS, which offers robustness guarantees for a model under a $L_2$-norm ball.

In particular, we follow the similar setting of Carlini et al. (2022) which uses a DDPM-based diffusion model. The overall algorithm contains three steps:

(1) Our framework estimates $n$, the number of steps used for the reverse process of DDPM-based diffusion model. Since Randomized Smoothing (Cohen et al., 2019) adds Gaussian noise $\boldsymbol{\epsilon}$, where $\boldsymbol{\epsilon} \sim \mathcal{N}(\mathbf{0}, \sigma^2 \boldsymbol{I})$, to data input $\boldsymbol{x}$ to get the randomized data input, $\boldsymbol{x}_{\mathrm{rs}} = \boldsymbol{x} + \boldsymbol{\epsilon}$, we map between the noise required by the randomized example $\boldsymbol{x}_{\mathrm{rs}}$ and the noise required by the diffused data $\boldsymbol{x}_n$ (i.e., $\boldsymbol{x}_n \sim \mathcal{N}(\boldsymbol{x}_n; \sqrt{\overline{\alpha}_n}\boldsymbol{x}_0, (1 - \overline{\alpha}_n)\boldsymbol{I})$) with $n$ step diffusion processing so that $\overline{\alpha}_n = \frac{1}{1+\sigma^2}$. In this way, we can compute the corresponding timestep $n$, where $n = \arg\min_s\{|\overline{\alpha}_s - \frac{1}{1+\sigma^2}| \mid s \in [N]\}$.

(2). Given the above calculated timestep $n$, we scale $\boldsymbol{x}_{rs}$ with $\sqrt{\overline{\alpha}_n}$ to obtain the scaled randomized smoothing sample $\sqrt{\overline{\alpha}_n}\boldsymbol{x}_{rs}$. Then we feed $\sqrt{\overline{\alpha}_n}\boldsymbol{x}_{rs}$ into the reverse process of the diffusion model by $K$-times to get the reversed sample set $\{\hat{\boldsymbol{x}}_0^1, \hat{\boldsymbol{x}}_0^2, \cdots, \hat{\boldsymbol{x}}_0^i, \cdots, \hat{\boldsymbol{x}}_0^K\}$.

(3). We feed the obtained reversed sample set into a standard *off-the-shelf* classifier $f$ to get the corresponding predicted labels $\{f(\hat{\boldsymbol{x}}_0^1), f(\hat{\boldsymbol{x}}_0^2), \ldots, f(\hat{\boldsymbol{x}}_0^i), \ldots, f(\hat{\boldsymbol{x}}_0^K)\}$, and apply *majority vote*, denoted $\mathbf{MV}(\cdots)$, on these predicted labels to get the final label for $\boldsymbol{x}_{rs}$.

**Fast Sampling.** To calculate the reversed sample, the standard reverse process of DDPM-based models require repeatedly applying a "single-step" operation $n$ times to get the reversed sample $\hat{\boldsymbol{x}}_0$ (i.e., $\hat{\boldsymbol{x}}_0 = \mathbf{Reverse}(\cdots \mathbf{Reverse}(\cdots \mathbf{Reverse}(\mathbf{Reverse}(\sqrt{\overline{\alpha}_n}\boldsymbol{x}_{rs}; n); n-1); \cdots; i); \cdots 1))$. Here $\hat{\boldsymbol{x}}_{i-1} = \mathbf{Reverse}(\hat{\boldsymbol{x}}_i; i)$ is equivalent to sample $\hat{\boldsymbol{x}}_{i-1}$ from $\mathcal{N}(\hat{\boldsymbol{x}}_{i-1}; \boldsymbol{\mu}_{\boldsymbol{\theta}}(\hat{\boldsymbol{x}}_i, i), \boldsymbol{\Sigma}_{\boldsymbol{\theta}}(\hat{\boldsymbol{x}}_i, i))$, where $\boldsymbol{\mu}_{\boldsymbol{\theta}}(\hat{\boldsymbol{x}}_i, i) = \frac{1}{\sqrt{1-\beta_i}}\left(\hat{\boldsymbol{x}}_i - \frac{\beta_i}{\sqrt{1-\overline{\alpha}_i}}\boldsymbol{\epsilon}_{\boldsymbol{\theta}}(\hat{\boldsymbol{x}}_i, i)\right)$ and $\boldsymbol{\Sigma}_{\boldsymbol{\theta}} := \exp(v \log \beta_i + (1 - v) \log \widetilde{\beta}_i)$. Here $v$ is a parameter learned by DDPM and $\widetilde{\beta}_i = \frac{1-\overline{\alpha}_{i-1}}{1-\overline{\alpha}_i}$.

To reduce the time complexity, we use the uniform sub-sampling strategy from Nichol & Dhariwal (2021). We uniformly sample a subsequence with size $b$ from the original $N$-step the reverse process. Note that Carlini et al. (2022) set $b = 1$ for the "one-shot" sampling, in this way, $\hat{\boldsymbol{x}}_0 = \frac{1}{\sqrt{\overline{\alpha}_n}}(\boldsymbol{x}_n - \sqrt{1 - \overline{\alpha}_n}\boldsymbol{\epsilon}_{\boldsymbol{\theta}}(\sqrt{\overline{\alpha}_n}\boldsymbol{x}_{rs}, n))$ is a deterministic value so that the reverse process does not obtain a posterior data distribution conditioned on the input. Instead, we can tune the number of the sub-sampled DDPM steps to be larger than one ($b > 1$) to sample from a posterior data distribution conditioned on the input. The details about the fast sampling are shown in appendix C.2.

| | | | Certified Accuracy at $\epsilon$(%) | | | | | | | |
| | | | CIFAR-10 | | | | ImageNet | | | |
| Method | Off-the-shelf | 0.25 | 0.5 | 0.75 | 1.0 | 0.5 | 1.0 | 1.5 | 2.0 | 3.0 |
|---|---|---|---|---|---|---|---|---|---|---|
| PixelDP (Lecuyer et al., 2019) | ✗ | $^{(71.0)}$22.0 | $^{(44.0)}$2.0 | - | - | $^{(33.0)}$16.0 | - | - | - | - |
| RS (Cohen et al., 2019) | ✗ | $^{(75.0)}$61.0 | $^{(75.0)}$43.0 | $^{(65.0)}$32.0 | $^{(65.0)}$23.0 | $^{(67.0)}$49.0 | $^{(57.0)}$37.0 | $^{(57.0)}$29.0 | $^{(44.0)}$19.0 | $^{(44.0)}$12.0 |
| SmoothAdv (Salman et al., 2019a) | ✗ | $^{(82.0)}$68.0 | $^{(76.0)}$54.0 | $^{(68.0)}$41.0 | $^{(64.0)}$32.0 | $^{(63.0)}$54.0 | $^{(56.0)}$42.0 | $^{(56.0)}$34.0 | $^{(41.0)}$26.0 | $^{(41.0)}$18.0 |
| Consistency (Jeong & Shin, 2020) | ✗ | $^{(77.8)}$68.8 | $^{(75.8)}$58.1 | $^{(72.9)}$48.5 | $^{(52.3)}$37.8 | $^{(55.0)}$50.0 | $^{(55.0)}$44.0 | $^{(55.0)}$34.0 | $^{(41.0)}$24.0 | $^{(41.0)}$17.0 |
| MACER (Zhai et al., 2020) | ✗ | $^{(81.0)}$71.0 | $^{(81.0)}$59.0 | $^{(66.0)}$46.0 | $^{(66.0)}$38.0 | $^{(68.0)}$57.0 | $^{(64.0)}$43.0 | $^{(64.0)}$31.0 | $^{(48.0)}$25.0 | $^{(48.0)}$14.0 |
| Boosting (Horváth et al., 2021) | ✗ | $^{(83.4)}$70.6 | $^{(76.8)}$60.4 | $^{(71.6)}$**52.4** | $^{(73.0)}$**38.8** | $^{(65.6)}$57.0 | $^{(57.0)}$44.6 | $^{(57.0)}$38.4 | $^{(44.6)}$28.6 | $^{(38.6)}$21.2 |
| SmoothMix (Jeong et al., 2021) | ✓ | $^{(77.1)}$67.9 | $^{(77.1)}$57.9 | $^{(74.2)}$47.7 | $^{(61.8)}$37.2 | $^{(55.0)}$50.0 | $^{(55.0)}$43.0 | $^{(55.0)}$38.0 | $^{(40.0)}$26.0 | $^{(40.0)}$17.0 |
| Denoised (Salman et al., 2020) | ✓ | $^{(72.0)}$56.0 | $^{(62.0)}$41.0 | $^{(62.0)}$28.0 | $^{(44.0)}$19.0 | $^{(60.0)}$33.0 | $^{(38.0)}$14.0 | $^{(38.0)}$6.0 | - | - |
| Lee (Lee, 2021) | ✓ | 60.0 | 42.0 | 28.0 | 19.0 | 41.0 | 24.0 | 11.0 | - | - |
| Carlini (Carlini et al., 2022) | ✓ | $^{(88.0)}$73.8 | $^{(88.0)}$56.2 | $^{(88.0)}$41.6 | $^{(74.2)}$31.0 | $^{(77.0)}$71.0 | $^{(74.0)}$54.0 | $^{(74.0)}$46.0 | $^{(59.0)}$29.0 | $^{(59.0)}$22.0 |
| **Ours** | ✓ | $^{(87.6)}$**76.6** | $^{(87.6)}$**64.6** | $^{(87.6)}$50.4 | $^{(73.6)}$37.4 | $^{(80.0)}$**76.0** | $^{(75.0)}$**62.0** | $^{(75.0)}$**49.0** | $^{(61.0)}$**37.0** | $^{(61.0)}$**26.0** |

Table 1: Certified accuracy compared with existing works. The certified accuracy at $\epsilon = 0$ for each model is in the parentheses. The certified accuracy for each cell is from the respective papers except Carlini et al. (2022). Our diffusion model and classifier are the same as Carlini et al. (2022), where the off-the-shelf classifier uses ViT-based architectures trained on a large dataset (ImageNet-22k).

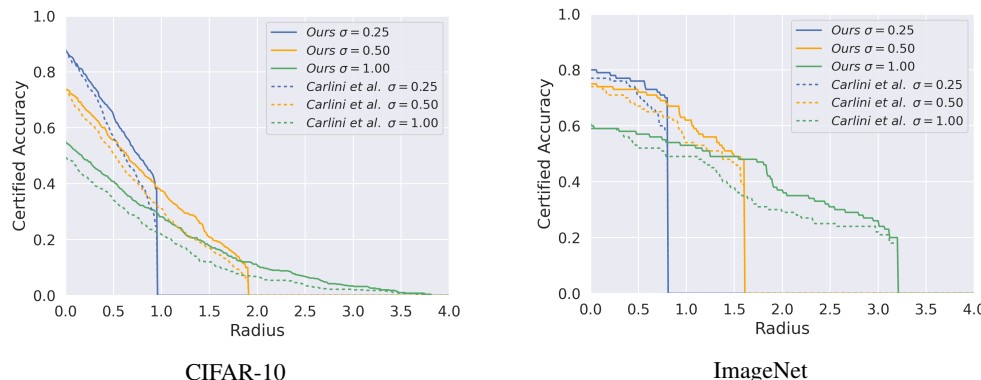

CIFAR-10                    ImageNet

Figure 3: Comparing our method vs Carlini et al. (2022) on CIFAR-10 and ImageNet. The lines represent the certified accuracy with different $L_2$ perturbation bound with different Gaussian noise $\sigma \in \{0.25, 0.50, 1.00\}$.

## 5 EXPERIMENTS

In this section, we use DensePure to evaluate certified robustness on two standard datasets, CIFAR-10 (Krizhevsky et al., 2009) and ImageNet (Deng et al., 2009).

**Experimental settings** We follow the experimental setting from Carlini et al. (2022). Specifically, for CIFAR-10, we use the 50-M unconditional improved diffusion model from Nichol & Dhariwal (2021) as the diffusion model. We select ViT-B/16 model Dosovitskiy et al. (2020) pretrained on ImageNet-21k and finetuned on CIFAR-10 as the classifier, which could achieve 97.9% accuracy on CIFAR-10. For ImageNet, we use the unconditional $256 \times 256$ guided diffusion model from Dhariwal & Nichol (2021) as the diffusion model and pretrained BEiT large model (Bao et al., 2021) trained on ImageNet-21k as the classifier, which could achieve 88.6% top-1 accuracy on validation set of ImageNet-1k. We select three different noise levels $\sigma \in \{0.25, 0.5, 1.0\}$ for certification. For the parameters of DensePure , we set $K = 40$ and $b = 10$ except the results in ablation study. The details about the baselines are in the appendix.

### 5.1 MAIN RESULTS

We perform DensePure on the subset of CIFAR-10 or ImageNet. We choose the same subset as in Cohen et al. (2019), 500 samples for CIFAR-10 and 100 samples for ImageNet ( the results with 500 samples are shown in the appendix D.10). The results are shown in Table 1. For CIFAR-10, comparing with the models which are *carefully* trained with randomized smoothing techniques in an end-to-end manner (i.e., w/o off-the-shelf classifier), we observe that our method with the standard off-the-shelf classifier outperforms them at smaller $\epsilon = \{0.25, 0.5\}$ on both CIFAR-10 and ImageNet datasets while achieves comparable performance at larger $\epsilon = \{0.75, 1.0\}$. Comparing with the non-diffusion model based methods with off-the-shelf classifier (i.e., Denoised (Salman et al., 2020) and Lee (Lee, 2021)), both our method and Carlini et al. (2022) are significantly better

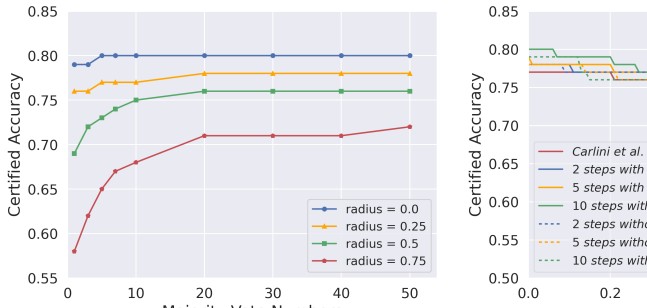

Figure 4: Ablation study on ImageNet. The left image shows the certified accuracy among different vote numbers with different radius $\epsilon \in \{0.0, 0.25, 0.5, 0.75\}$. The right image shows the certified accuracy with different fast sampling steps $b$.

than them. These results verify the non-trivial adversarial robustness improvements introduced from the diffusion model. For ImageNet, our method is consistently better than all priors with a large margin.

Since both Carlini et al. (2022) and DensePure use the diffusion model, to better understand the importance of our design, that approximates the label of the high density region in the conditional distribution, we compare DensePure with Carlini et al. (2022) in a more fine-grained manner.

We show detailed certified robustness of the model among different $\sigma$ at different radius for CIFAR-10 in Figure 3-left and for ImageNet in Figure 3-right. We also present our results of certified accuracy at different $\epsilon$ in Appendix D.3. From these results, we find that our method is still consistently better at most $\epsilon$ (except $\epsilon = 0$) among different $\sigma$. The performance margin between ours and Carlini et al. (2022) will become even larger with a large $\epsilon$. These results further indicate that although the diffusion model improves model robustness, leveraging the posterior data distribution conditioned on the input instance (like DensePure ) via reverse process instead of using single sample ((Carlini et al., 2022)) is the key for better robustness. Additionally, we use the off-the-shelf classifiers, which are the VIT-based architectures trained a larger dataset. In the later ablation study section, we select the CNN-based architecture wide-ResNet trained on standard dataset from scratch. Our method still achieves non-trivial robustness. Further, our experiments in Appendix D.7 shows that removing the diffusion model from DensePure deteriorates the performance. It further verifies that our design is non-trivial.

## 5.2 ABLATION STUDY

**Voting samples ($K$)** We first show how $K$ affects the certified accuracy. For efficiency, we select $b = 10$. We conduct experiments for both datasets. We show the certified accuracy among different $r$ at $\sigma = 0.25$ in Figure 4. The results for $\sigma = 0.5, 1.0$ and CIFAR-10 are shown in the Appendix D.4. Comparing with the baseline (Carlini et al., 2022), we find that a larger majority vote number leads to a better certified accuracy. It verifies that DensePure indeed benefits the adversarial robustness and making a good approximation of the label with high density region requires a large number of voting samples. We find that our certified accuracy will almost converge at $r = 40$. Thus, we set $r = 40$ for our experiments. The results with other $\sigma$ show the similar tendency. To further improve the time efficiency, we can use $K$-Consensus (Horváth et al., 2021). It accelerates the majority vote process by $45\% \sim 60\%$ with a negligible performance drop. The experimental details and results are in Appendix D.8.

**Fast sampling steps ($b$)** To investigate the role of $b$, we conduct additional experiments with $b \in \{2, 5\}$ at $\sigma = 0.25$. The results on ImageNet are shown in Figure 4 and results for $\sigma = 0.5, 1.0$ and CIFAR-10 are shown in the Appendix D.5. By observing results *with* majority vote, we find that a larger $b$ can lead to a better certified accuracy since a larger $b$ generates images with higher quality. By observing results *without* majority vote, the results show opposite conclusions where a larger $b$ leads to a lower certified accuracy, which contradicts to our intuition. We guess the potential reason is that though more sampling steps can normally lead to better image recovery quality, it also brings more randomness, increasing the probability that the reversed image locates into a data region with the wrong label. These results further verify that majority vote is necessary for a better performance.

| Datasets | Methods | Model | Certified Accuracy at $\epsilon(\%)$ | | | | | | | | |
|---|---|---|---|---|---|---|---|---|---|---|---|
| | | | 0.0 | 0.25 | 0.5 | 0.75 | Model | 0.0 | 0.25 | 0.5 | 0.75 |
| CIFAR-10 | Carlini (Carlini et al., 2022) | ViT-B/16 | **93.0** | 76.0 | 57.0 | 47.0 | WRN28-10 | 86.0 | 66.0 | 55.0 | 37.0 |
| | **Ours** | ViT-B/16 | 92.0 | **82.0** | **69.0** | **56.0** | WRN28-10 | **90.0** | **77.0** | **63.0** | **50.0** |
| ImageNet | Carlini (Carlini et al., 2022) | BEiT | 77.0 | 76.0 | 71.0 | 60.0 | WRN50-2 | 73.0 | 67.0 | 57.0 | 48.0 |
| | **Ours** | BEiT | **80.0** | **78.0** | **76.0** | **71.0** | WRN50-2 | **81.0** | **72.0** | **66.0** | **61.0** |

Table 2: Certified accuracy of our method among different classifier. BeiT and ViT are pre-trained on a larger dataset ImageNet-22k and fine-tuned at ImageNet-1k and CIFAR-10 respectively. WideResNet is trained on ImageNet-1k for ImageNet and trained on CIFAR-10 from scratch for CIFAR-10.

**Different architectures** One advantage of DensePure is to use the off-the-shelf classifier so that it can plug in any classifier. We choose Convolutional neural network (CNN)-based architectures: Wide-ResNet28-10 (Zagoruyko & Komodakis, 2016) for CIFAR-10 with $95.1\%$ accuracy and Wide-ResNet50-2 for ImageNet with $81.5\%$ top-1 accuracy, at $\sigma = 0.25$. The results are shown in Table 2 and Figure E in Appendix D.6. Results for more model architectures and $\sigma$ of ImageNet are also shown in Appendix D.6. We show that our method can enhance the certified robustness of any given classifier trained on the original data distribution. Noticeably, although the performance of CNN-based classifier is lower than Transformer-based classifier, DensePure with CNN-based model as the classifier can outperform Carlini et al. (2022) with ViT-based model as the classifier (except $\epsilon = 0$ for CIFAR-10).

## 6 RELATED WORK

Using an off-the-shelf generative model to purify adversarial perturbations has become an important direction in adversarial defense. Previous works have developed various purification methods based on different generative models, such as GANs (Samangouei et al., 2018), autoregressive generative models (Song et al., 2018), and energy-based models (Du & Mordatch, 2019; Grathwohl et al., 2020; Hill et al., 2021). More recently, as diffusion models (or score-based models) achieve better generation quality than other generative models (Ho et al., 2020; Dhariwal & Nichol, 2021), many works consider using diffusion models for adversarial purification (Nie et al., 2022; Wu et al., 2022; Sun et al., 2022) Although they have found good empirical results in defending against existing adversarial attacks (Nie et al., 2022), there is no provable guarantee about the robustness about such methods. On the other hand, certified defenses provide guarantees of robustness (Mirman et al., 2018; Cohen et al., 2019; Lecuyer et al., 2019; Salman et al., 2020; Horváth et al., 2021; Zhang et al., 2018; Raghunathan et al., 2018a;b; Salman et al., 2019b; Wang et al., 2021). They provide a lower bounder of model accuracy under constrained perturbations. Among them, approaches Lecuyer et al. (2019); Cohen et al. (2019); Salman et al. (2019a); Jeong & Shin (2020); Zhai et al. (2020); Horváth et al. (2021); Jeong et al. (2021); Salman et al. (2020); Lee (2021); Carlini et al. (2022) based on randomized smoothing (Cohen et al., 2019) show the great scalability and achieve promising performance on large network and dataset. The most similar work to us is Carlini et al. (2022), which uses diffusion models combined with standard classifiers for certified defense. They view diffusion model as blackbox without having a theoretical under- standing of why and how the diffusion models contribute to such nontrivial certified robustness.

## 7 CONCLUSION

In this work, we theoretically prove that the diffusion model could purify adversarial examples back to the corresponding clean sample with high probability, as long as the data density of the corresponding clean samples is high enough. Our theoretical analysis characterizes the conditional distribution of the reversed samples given the adversarial input, generated by the diffusion model reverse process. Using the highest density point in the conditional distribution as the deterministic reversed sample, we identify the robust region of a given instance under the diffusion model reverse process, which is potentially much larger than previous methods. Our analysis inspires us to propose an effective pipeline DensePure, for adversarial robustness. We conduct comprehensive experiments to show the effectiveness of DensePure by evaluating the certified robustness via the randomized smoothing algorithm. Note that DensePure is an off-the-shelf pipeline that does not require training a smooth classifier. Our results show that DensePure achieves the new SOTA certified robustness for perturbation with $\mathcal{L}_2$-norm. We hope that our work sheds light on an in-depth understanding of the diffusion model for adversarial robustness.

**Limitations.** The time complexity of DensePure is high since it requires repeating the reverse process multiple times. In this paper, we use fast sampling to reduce the time complexity and show that the setting ($b = 2$ and $K = 10$) can achieve nontrivial certified accuracy. We leave the more advanced fast sampling strategy as the future direction.

## ETHICS STATEMENT

Our work can positively impact the society by improving the robustness and security of AI systems. We have not involved human subjects or data set releases; instead, we carefully follow the provided licenses of existing data and models for developing and evaluating our method.

## 8 ACKNOWLEDGMENT

We thank the support of NSF grant No.1910100, NSF CNS 2046726, C3 AI and DHS under grant No. 17STQAC00001-06-00, DARPA under grant N66001-15-C-4066, the Center for Long-Term Cybersecurity, and Berkeley Deep Drive. Any opinions, findings, conclusions, or recommendations expressed in this material are those of the authors, and do not necessarily reflect the views of the sponsors.

## REPRODUCIBILITY STATEMENT

For theoretical analysis, all necessary assumptions are listed in B.1 and the complete proofs are included in B.2. The experimental setting and datasets are provided in section 5. The pseudo-code for DensePure is in C.1 and the fast sampling procedures are provided in C.2.

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

# APPENDIX

Here is the appendix.

## A NOTATIONS

| | |
|---|---|
| $p$ | data distribution |
| $\mathbb{P}(A)$ | probability of event $A$ |
| $\mathcal{C}^k$ | set of functions with continuous $k$-th derivatives |
| $\boldsymbol{w}(t)$ | standard Wiener Process |
| $\overline{\boldsymbol{w}}(t)$ | reverse-time standard Wiener Process |
| $h(\boldsymbol{x}, t)$ | drift coefficient in SDE |
| $g(t)$ | diffusion coefficient in SDE |
| $\alpha_t$ | scaling coefficient at time $t$ |
| $\sigma_t^2$ | variance of added Gaussian noise at time $t$ |
| $\{\mathbf{x}_t\}_{t \in [0,1]}$ | diffusion process generated by SDE |
| $\{\hat{\mathbf{x}}_t\}_{t \in [0,1]}$ | reverse process generated by reverse-SDE |
| $p_t$ | distribution of $\mathbf{x}_t$ and $\hat{\mathbf{x}}_t$ |
| $\{\mathbf{x}_1, \mathbf{x}_2, \ldots, \mathbf{x}_N\}$ | diffusion process generated by DDPM |
| $\{\beta_i\}_{i=1}^N$ | pre-defined noise scales in DDPM |
| $\boldsymbol{\epsilon}_a$ | adversarial attack |
| $\boldsymbol{x}_a$ | adversarial sample |
| $\boldsymbol{x}_{a,t}$ | scaled adversarial sample |
| $f(\cdot)$ | classifier |
| $g(\cdot)$ | smoothed classifier |
| $\mathbb{P}\left(\hat{\mathbf{x}}_0 = \boldsymbol{x} \mid \hat{\mathbf{x}}_t = \boldsymbol{x}_{a,t}\right)$ | density of conditional distribution generated by reverse-SDE based on $\boldsymbol{x}_{a,t}$ |
| $\mathcal{P}(\boldsymbol{x}_a; t)$ | purification model with highest density point |
| $\mathcal{G}(\boldsymbol{x}_0)$ | data region with the same label as $\boldsymbol{x}_0$ |
| $\mathcal{D}_{\mathcal{P}}^f(\mathcal{G}(\boldsymbol{x}_0); t)$ | robust region for $\mathcal{G}(\boldsymbol{x}_0)$ associated with base classifier $f$ and purification model $\mathcal{P}$ |
| $r_{\mathcal{P}}^f(\boldsymbol{x}_0; t)$ | robust radius for the point associated with base classifier $f$ and purification model $\mathcal{P}$ |
| $\mathcal{D}_{sub}(\boldsymbol{x}_0; t)$ | convex robust sub-region |
| $\boldsymbol{s}_\theta(\boldsymbol{x}, t)$ | score function |
| $\{\mathbf{x}_t^\theta\}_{t \in [0,1]}$ | reverse process generated by score-based diffusion model |
| $\mathbb{P}\left(\mathbf{x}_0^\theta = \boldsymbol{x} \mid \mathbf{x}_t^\theta = \boldsymbol{x}_{a,t}\right)$ | density of conditional distribution generated by score-based diffusion model based on $\boldsymbol{x}_{a,t}$ |
| $\lambda(\tau)$ | weighting scheme of training loss for score-based diffusion model |
| $\mathcal{J}_{\mathrm{SM}}(\theta, t; \lambda(\cdot))$ | truncated training loss for score-based diffusion model |
| $\boldsymbol{\mu}_t, \boldsymbol{\nu}_t$ | path measure for $\{\hat{\mathbf{x}}_\tau\}_{\tau \in [0,t]}$ and $\{\mathbf{x}_\tau^\theta\}_{\tau \in [0,t]}$ respectively |

## B MORE DETAILS ABOUT THEORETICAL ANALYSIS

### B.1 ASSUMPTIONS

(i) The data distribution $p \in \mathcal{C}^2$ and $\mathbb{E}_{\boldsymbol{x} \sim p}[||\boldsymbol{x}||_2^2] < \infty$.

(ii) $\forall t \in [0, T] : h(\cdot, t) \in \mathcal{C}^1, \exists C > 0, \forall \boldsymbol{x} \in \mathbb{R}^n, t \in [0, T] : ||h(\boldsymbol{x}, t)||_2 \leqslant C(1 + ||\boldsymbol{x}||_2)$.

(iii) $\exists C > 0, \forall \boldsymbol{x}, \boldsymbol{y} \in \mathbb{R}^n : ||h(\boldsymbol{x}, t) - h(\boldsymbol{y}, t)||_2 \leqslant C||\boldsymbol{x} - \boldsymbol{y}||_2$.

(iv) $g \in \mathcal{C}$ and $\forall t \in [0, T], |g(t)| > 0$.

(v) $\forall t \in [0, T] : \boldsymbol{s}_\theta(\cdot, t) \in \mathcal{C}^1, \exists C > 0, \forall \boldsymbol{x} \in \mathbb{R}^n, t \in [0, T] : ||\boldsymbol{s}_\theta(\boldsymbol{x}, t)||_2 \leqslant C(1 + ||\boldsymbol{x}||_2)$.

(vi) $\exists C > 0, \forall \boldsymbol{x}, \boldsymbol{y} \in \mathbb{R}^n : ||\boldsymbol{s}_\theta(\boldsymbol{x}, t) - \boldsymbol{s}_\theta(\boldsymbol{y}, t)||_2 \leqslant C||\boldsymbol{x} - \boldsymbol{y}||_2$.

### B.2 THEOREMS AND PROOFS

**Theorem 3.1.** *Under conditions B.1, solving equation reverse-SDE starting from time $t$ and point $\boldsymbol{x}_{a,t} = \sqrt{\alpha_t}\boldsymbol{x}_a$ will generate a reversed random variable $\hat{\mathbf{x}}_0$ with conditional distribution*

$$\mathbb{P}\left(\hat{\mathbf{x}}_0 = \boldsymbol{x} | \hat{\mathbf{x}}_t = \boldsymbol{x}_{a,t}\right) \propto p(\boldsymbol{x}) \cdot \frac{1}{\sqrt{(2\pi\sigma_t^2)^n}} e^{\frac{-||\boldsymbol{x} - \boldsymbol{x}_a||_2^2}{2\sigma_t^2}}$$

*where $\sigma_t^2 = \frac{1 - \alpha_t}{\alpha_t}$ is the variance of the Gaussian noise added at timestamp $t$ in the diffusion process SDE.*

*Proof.* Under the assumption, we know $\{\mathbf{x}_t\}_{t \in [0,1]}$ and $\{\hat{\mathbf{x}}_t\}_{t \in [0,1]}$ follow the same distribution, which means

$$
\begin{aligned}
\mathbb{P}\left(\hat{\mathbf{x}}_0 = \boldsymbol{x} | \hat{\mathbf{x}}_t = \boldsymbol{x}_{a,t}\right) &= \frac{\mathbb{P}(\hat{\mathbf{x}}_0 = \boldsymbol{x}, \hat{\mathbf{x}}_t = \boldsymbol{x}_{a,t})}{\mathbb{P}(\hat{\mathbf{x}}_t = \boldsymbol{x}_{a,t})} \\
&= \frac{\mathbb{P}(\mathbf{x}_0 = \boldsymbol{x}, \mathbf{x}_t = \boldsymbol{x}_{a,t})}{\mathbb{P}(\mathbf{x}_t = \boldsymbol{x}_{a,t})} \\
&= \mathbb{P}(\mathbf{x}_0 = \boldsymbol{x}) \frac{\mathbb{P}(\mathbf{x}_t = \boldsymbol{x}_{a,t} | \mathbf{x}_0 = \boldsymbol{x})}{\mathbb{P}(\mathbf{x}_t = \boldsymbol{x}_{a,t})} \\
&\propto \mathbb{P}(\mathbf{x}_0 = \boldsymbol{x}) \frac{1}{\sqrt{(2\pi\sigma_t^2)^n}} e^{\frac{-||\boldsymbol{x} - \boldsymbol{x}_a||_2^2}{2\sigma_t^2}} \\
&= p(\boldsymbol{x}) \cdot \frac{1}{\sqrt{(2\pi\sigma_t^2)^n}} e^{\frac{-||\boldsymbol{x} - \boldsymbol{x}_a||_2^2}{2\sigma_t^2}}
\end{aligned}
$$

where the third equation is due to the chain rule of probability and the last equation is a result of the diffusion process. $\square$

**Theorem 3.3.** *Under conditions B.1 and classifier $f$, let $\boldsymbol{x}_0$ be the sample with ground-truth label and $\boldsymbol{x}_a$ be the adversarial sample, then (i) the purified sample $\mathcal{P}(\boldsymbol{x}_a; t)$ will have the ground-truth label if $\boldsymbol{x}_a$ falls into the following convex set,*

$$\mathcal{D}_{sub}(\boldsymbol{x}_0; t) := \bigcap_{\{\boldsymbol{x}_0': f(\boldsymbol{x}_0') \neq f(\boldsymbol{x}_0)\}} \left\{ \boldsymbol{x}_a : (\boldsymbol{x}_a - \boldsymbol{x}_0)^\top (\boldsymbol{x}_0' - \boldsymbol{x}_0) < \sigma_t^2 \log\left(\frac{p(\boldsymbol{x}_0)}{p(\boldsymbol{x}_0')}\right) + \frac{||\boldsymbol{x}_0' - \boldsymbol{x}_0||_2^2}{2} \right\},$$

*and further, (ii) the purified sample $\mathcal{P}(\boldsymbol{x}_a; t)$ will have the ground-truth label if and only if $\boldsymbol{x}_a$ falls into the following set, $\mathcal{D}(\mathcal{G}(\boldsymbol{x}_0); t) := \bigcup_{\tilde{\boldsymbol{x}}_0 : f(\tilde{\boldsymbol{x}}_0) = f(\boldsymbol{x}_0)} \mathcal{D}_{sub}(\tilde{\boldsymbol{x}}_0; t)$. In other words, $\mathcal{D}(\mathcal{G}(\boldsymbol{x}_0); t)$ is the robust region for data region $\mathcal{G}(\boldsymbol{x}_0)$ under $\mathcal{P}(\cdot; t)$ and $f$.*

*Proof.* We start with part (i).

The main idea is to prove that a point $\boldsymbol{x}_0'$ such that $f(\boldsymbol{x}_0') \neq f(\boldsymbol{x}_0)$ should have lower density than $\boldsymbol{x}_0$ in the conditional distribution in Theorem 3.1 so that $\mathcal{P}(\boldsymbol{x}_a; t)$ cannot be $\boldsymbol{x}_0'$. In other words, we should have

$$\mathbb{P}(\hat{\mathbf{x}}_0 = \boldsymbol{x}_0 | \hat{\mathbf{x}}_t = \boldsymbol{x}_{a,t}) > \mathbb{P}(\hat{\mathbf{x}}_0 = \boldsymbol{x}_0' \mid \hat{\mathbf{x}}_t = \boldsymbol{x}_{a,t}).$$

By Theorem 3.1, this is equivalent to

$$p(\boldsymbol{x}_0) \cdot \frac{1}{\sqrt{(2\pi\sigma_t^2)^n}} e^{\frac{-||\boldsymbol{x}_0 - \boldsymbol{x}_a||_2^2}{2\sigma_t^2}} > p(\boldsymbol{x}_0') \cdot \frac{1}{\sqrt{(2\pi\sigma_t^2)^n}} e^{\frac{-||\boldsymbol{x}_0' - \boldsymbol{x}_a||_2^2}{2\sigma_t^2}}$$

$$\Leftrightarrow \log\left(\frac{p(\boldsymbol{x}_0)}{p(\boldsymbol{x}_0')}\right) > \frac{1}{2\sigma_t^2}\left(||\boldsymbol{x}_0 - \boldsymbol{x}_a||_2^2 - ||\boldsymbol{x}_0' - \boldsymbol{x}_a||_2^2\right)$$

$$\Leftrightarrow \log\left(\frac{p(\boldsymbol{x}_0)}{p(\boldsymbol{x}_0')}\right) > \frac{1}{2\sigma_t^2}\left(||\boldsymbol{x}_0 - \boldsymbol{x}_a||_2^2 - ||\boldsymbol{x}_0' - \boldsymbol{x}_0 + \boldsymbol{x}_0 - \boldsymbol{x}_a||_2^2\right)$$

$$\Leftrightarrow \log\left(\frac{p(\boldsymbol{x}_0)}{p(\boldsymbol{x}_0')}\right) > \frac{1}{2\sigma_t^2}\left(2(\boldsymbol{x}_a - \boldsymbol{x}_0)^\top(\boldsymbol{x}_0' - \boldsymbol{x}_0) - ||\boldsymbol{x}_0' - \boldsymbol{x}_0||_2^2\right).$$

Re-organizing the above inequality, we obtain

$$(\boldsymbol{x}_a - \boldsymbol{x}_0)^\top(\boldsymbol{x}_0' - \boldsymbol{x}_0) < \sigma_t^2 \log\left(\frac{p(\boldsymbol{x}_0)}{p(\boldsymbol{x}_0')}\right) + \frac{1}{2}||\boldsymbol{x}_0' - \boldsymbol{x}_0||_2^2.$$

Note that the order of $\boldsymbol{x}_a$ is at most one in every term of the above inequality, so the inequality actually defines a half-space in $\mathbb{R}^n$ for every $(\boldsymbol{x}_0, \boldsymbol{x}_0')$ pair. Further, we have to satisfy the inequality for every $\boldsymbol{x}_0'$ such that $f(\boldsymbol{x}_0') \neq f(\boldsymbol{x}_0)$, therefore, by intersecting over all such half-spaces, we obtain a convex $\mathcal{D}_{\text{sub}}(\boldsymbol{x}_0; t)$.

Then we prove part (ii).

On the one hand, if $\boldsymbol{x}_a \in \mathcal{D}(\mathcal{G}(\boldsymbol{x}_0); t)$, then there exists one $\tilde{\boldsymbol{x}}_0$ such that $f(\tilde{\boldsymbol{x}}_0) = f(\boldsymbol{x}_0)$ and $\boldsymbol{x}_a \in \mathcal{D}_{\text{sub}}(\tilde{\boldsymbol{x}}_0; t)$. By part (i), $\tilde{\boldsymbol{x}}_0$ has higher probability than all other points with different labels from $\boldsymbol{x}_0$ in the conditional distribution $\mathbb{P}(\hat{\mathbf{x}}_0 = \boldsymbol{x} | \hat{\mathbf{x}}_t = \boldsymbol{x}_{a,t})$ characterized by Theorem 3.1. Therefore, $\mathcal{P}(\boldsymbol{x}_a; t)$ should have the same label as $\boldsymbol{x}_0$. On the other hand, if $\boldsymbol{x}_a \notin \mathcal{D}(\mathcal{G}(\boldsymbol{x}_0); t)$, then there is a point $\tilde{\boldsymbol{x}}_1$ with different label from $\boldsymbol{x}_0$ such that for any $\tilde{\boldsymbol{x}}_0$ with the same label as $\boldsymbol{x}_0$, $\mathbb{P}(\hat{\mathbf{x}}_0 = \tilde{\boldsymbol{x}}_1 | \hat{\mathbf{x}}_t = \boldsymbol{x}_{a,t}) > \mathbb{P}(\hat{\mathbf{x}}_0 = \tilde{\boldsymbol{x}}_0 | \hat{\mathbf{x}}_t = \boldsymbol{x}_{a,t})$. In other words, $\mathcal{P}(\boldsymbol{x}_a; t)$ would have different label from $\boldsymbol{x}_0$. □

**Theorem 3.4.** *Under score-based diffusion model Song et al. (2021b) and conditions B.1, we can bound*

$$D_{KL}(\mathbb{P}(\hat{\mathbf{x}}_0 = \boldsymbol{x} \mid \hat{\mathbf{x}}_t = \boldsymbol{x}_{a,t}) \| \mathbb{P}(\mathbf{x}_0^\theta = \boldsymbol{x} \mid \mathbf{x}_t^\theta = \boldsymbol{x}_{a,t})) = \mathcal{J}_{\text{SM}}(\theta, t; \lambda(\cdot))$$

*where $\{\hat{\boldsymbol{x}}_\tau\}_{\tau \in [0,t]}$ and $\{\boldsymbol{x}_\tau^\theta\}_{\tau \in [0,t]}$ are stochastic processes generated by reverse-SDE and score-based diffusion model respectively,*

$$\mathcal{J}_{\text{SM}}(\theta, t; \lambda(\cdot)) := \frac{1}{2}\int_0^t \mathbb{E}_{p_\tau(\mathbf{x})}\left[\lambda(\tau)\|\nabla_{\mathbf{x}}\log p_\tau(\mathbf{x}) - \boldsymbol{s}_\theta(\mathbf{x}, \tau)\|_2^2\right]\mathrm{d}\tau,$$

*$\boldsymbol{s}_\theta(\mathbf{x}, \tau)$ is the score function to approximate $\nabla_{\mathbf{x}}\log p_\tau(\mathbf{x})$, and $\lambda : \mathbb{R} \to \mathbb{R}$ is any weighting scheme used in the training score-based diffusion models.*

*Proof.* Similar to proof of (Song et al., 2021a, Theorem 1), let $\boldsymbol{\mu}_t$ and $\boldsymbol{\nu}_t$ be the path measure for reverse processes $\{\hat{\mathbf{x}}_\tau\}_{\tau \in [0,t]}$ and $\{\mathbf{x}_\tau^\theta\}_{\tau \in [0,t]}$ respectively based on the scaled adversarial sample $\boldsymbol{x}_{a,t}$. Under conditions B.1, the KL-divergence can be computed via the Girsanov theorem Oksendal

(2013):

$$D_{\mathrm{KL}}\left(\mathbb{P}(\hat{\mathbf{x}}_0 = \boldsymbol{x} \mid \hat{\mathbf{x}}_t = \boldsymbol{x}_{a,t}) \| \mathbb{P}(\mathbf{x}_0^\theta = \boldsymbol{x} \mid \mathbf{x}_t^\theta = \boldsymbol{x}_{a,t})\right)$$

$$= -\mathbb{E}_{\boldsymbol{\mu}_t}\left[\log \frac{d\boldsymbol{\nu}_t}{d\boldsymbol{\mu}_t}\right]$$

$$\overset{(i)}{=} \mathbb{E}_{\boldsymbol{\mu}_t}\left[\int_0^t g(\tau)\left(\nabla_{\mathbf{x}} \log p_\tau(\mathbf{x}) - \boldsymbol{s}_\theta(\mathbf{x},\tau)\right) \mathrm{d}\overline{\mathbf{w}}_\tau + \frac{1}{2}\int_0^t g(\tau)^2 \left\|\nabla_{\mathbf{x}} \log p_\tau(\mathbf{x}) - \boldsymbol{s}_\theta(\mathbf{x},\tau)\right\|_2^2 \mathrm{d}\tau\right]$$

$$= \mathbb{E}_{\boldsymbol{\mu}_t}\left[\frac{1}{2}\int_0^t g(\tau)^2 \left\|\nabla_{\mathbf{x}} \log p_\tau(\mathbf{x}) - s_\theta(\mathbf{x},\tau)\right\|_2^2 \mathrm{d}\tau\right]$$

$$= \frac{1}{2}\int_0^\tau \mathbb{E}_{p_\tau(\mathbf{x})}\left[g(\tau)^2 \left\|\nabla_{\mathbf{x}} \log p_\tau(\mathbf{x}) - s_\theta(\mathbf{x},\tau)\right\|_2^2\right] \mathrm{d}\tau$$

$$= \mathcal{J}_{\mathrm{SM}}\left(\theta, t; g(\cdot)^2\right)$$

where (i) is due to Girsanov Theorem and (ii) is due to the martingale property of Itô integrals. $\qquad\square$

## C MORE DETAILS ABOUT DENSEPURE

### C.1 PSEUDO-CODE

We provide the pseudo code of DensePure in Algo. 1 and Alg. 2

---

**Algorithm 1** DensePure pseudo-code with the highest density point

---
1: Initialization: choose off-the-shelf diffusion model and classifier $f$, choose $\psi = t$,
2: Input sample $\boldsymbol{x}_a = \boldsymbol{x}_0 + \boldsymbol{\epsilon}_a$
3: Compute $\hat{\boldsymbol{x}}_0 = \mathcal{P}(\boldsymbol{x}_a; \psi)$
4: $\hat{y} = f(\hat{\boldsymbol{x}}_0)$

---

**Algorithm 2** DensePure pseudo-code with majority vote

---
1: Initialization: choose off-the-shelf diffusion model and classifier $f$, choose $\sigma$
2: Compute $\overline{\alpha}_n = \frac{1}{1+\sigma^2}$, $n = \arg\min_s\left\{\left|\overline{\alpha}_s - \frac{1}{1+\sigma^2}\right| \mid s \in \{1, 2, \cdots, N\}\right\}$
3: Generate input sample $\boldsymbol{x}_{\mathrm{rs}} = \boldsymbol{x}_0 + \boldsymbol{\epsilon}, \boldsymbol{\epsilon} \sim \mathcal{N}(\mathbf{0}, \sigma^2 \boldsymbol{I})$
4: Choose schedule $S^b$, get $\hat{\boldsymbol{x}}_0^i \leftarrow \mathbf{rev}(\sqrt{\overline{\alpha}_n} \boldsymbol{x}_{\mathrm{rs}})_i, i = 1, 2, \ldots, K$ with Fast Sampling
5: $\hat{y} = \mathbf{MV}(\{f(\hat{\boldsymbol{x}}_0^1), \ldots, f(\hat{\boldsymbol{x}}_0^K)\}) = \arg\max_c \sum_{i=1}^K \mathbf{1}\{f(\hat{\boldsymbol{x}}_0^i) = c\}$

---

### C.2 DETAILS ABOUT FAST SAMPLING

Applying single-step operation $n$ times is a time-consuming process. In order to reduce the time complexity, we follow the method used in (Nichol & Dhariwal, 2021) and sample a subsequence $S^b$ with $b$ values (i.e., $S^b = \underbrace{\{n, \lfloor n - \frac{n}{b}\rfloor, \cdots, 1\}}_b$, where $S_j^b$ is the $j$-th element in $S^b$ and $S_j^b = \lfloor n - \frac{jn}{b}\rfloor, \forall j < b$ and $S_b^b = 1$) from the original schedule $S$ (i.e., $S = \underbrace{\{n, n-1, \cdots, 1\}}_n$, where $S_j = j$ is the $j$-th element in $S$).

Within this context, we adapt the original $\overline{\alpha}$ schedule $\overline{\alpha}^S = \{\overline{\alpha}_1, \cdots, \overline{\alpha}_i, \cdots, \overline{\alpha}_n\}$ used for single-step to the new schedule $\overline{\alpha}^{S^b} = \{\overline{\alpha}_{S_1^b}, \cdots, \overline{\alpha}_{S_j^b}, \cdots, \overline{\alpha}_{S_b^b}\}$ (i.e., $\overline{\alpha}_i^{S^b} = \overline{\alpha}_{S_i^b} = \overline{\alpha}_{S_{\lfloor n - \frac{in}{b}\rfloor}}$ is the $i$-th element in $\overline{\alpha}^{S^b}$). We calculate the corresponding $\beta^{S^b} = \{\beta_1^{S^b}, \beta_2^{S^b}, \cdots, \beta_i^{S^b}, \cdots, \beta_b^{S^b}\}$ and $\widetilde{\beta}^{S^b} = \{\widetilde{\beta}_1^{S^b}, \widetilde{\beta}_2^{S^b}, \cdots, \widetilde{\beta}_i^{S^b}, \cdots, \widetilde{\beta}_b^{S^b}\}$ schedules, where $\beta_{S_i^b} = \beta_i^{S^b} = 1 - \frac{\overline{\alpha}_i^{S^b}}{\overline{\alpha}_{i-1}^{S^b}}$, $\widetilde{\beta}_{S_i^b} = \widetilde{\beta}_i^{S^b} = \frac{1 - \overline{\alpha}_{i-1}^{S^b}}{1 - \overline{\alpha}_i^{S^b}} \beta_{S_i^b}$. With these new schedules, we can use $b$ times reverse steps to calculate

| Methods | Noise | Certified Accuracy at $\epsilon(\%)$ | | | | |
|---|---|---|---|---|---|---|
| | | 0.0 | 0.25 | 0.5 | 0.75 | 1.0 |
| Carlini (Carlini et al., 2022) | $\sigma = 0.25$ | **88.0** | 73.8 | 56.2 | 41.6 | 0.0 |
| | $\sigma = 0.5$ | 74.2 | 62.0 | 50.4 | 40.2 | 31.0 |
| | $\sigma = 1.0$ | 49.4 | 41.4 | 34.2 | 27.8 | 21.8 |
| **Ours** | $\sigma = 0.25$ | 87.6(-0.4) | **76.6(+2.8)** | **64.6(+8.4)** | **50.4(+8.8)** | 0.0(+0.0) |
| | $\sigma = 0.5$ | 73.6(-0.6) | 65.4(+3.4) | 55.6(+5.2) | 46.0(+5.8) | **37.4(+6.4)** |
| | $\sigma = 1.0$ | 55.0(+5.6) | 47.8(+6.4) | 40.8(+6.6) | 33.0(+5.2) | 28.2(+6.4) |

Table A: Certified accuracy compared with Carlini et al. (2022) for CIFAR-10 at all $\sigma$. The numbers in the bracket are the difference of certified accuracy between two methods. Our diffusion model and classifier are the same as Carlini et al. (2022).

$\hat{\boldsymbol{x}}_0 = \underbrace{\textbf{Reverse}(\cdots \textbf{Reverse}(\textbf{Reverse}(\boldsymbol{x}_n; S_b^b); S_{b-1}^b); \cdots; 1)}_{b}$. Since $\boldsymbol{\Sigma}_{\boldsymbol{\theta}}(\boldsymbol{x}_{S_i^b}, S_i^b)$ is parameterized

as a range between $\beta^{S^b}$ and $\widetilde{\beta}^{S^b}$, it will automatically be rescaled. Thus, $\hat{\boldsymbol{x}}_{S_{i-1}^b} = \textbf{Reverse}(\hat{\boldsymbol{x}}_{S_i^b}; S_i^b)$ is equivalent to sample $\boldsymbol{x}_{S_{i-1}^b}$ from $\mathcal{N}(\boldsymbol{x}_{S_{i-1}^b}; \boldsymbol{\mu}_{\boldsymbol{\theta}}(\boldsymbol{x}_{S_i^b}, S_i^b), \boldsymbol{\Sigma}_{\boldsymbol{\theta}}(\boldsymbol{x}_{S_i^b}, S_i^b))$.

# D MORE EXPERIMENTAL DETAILS AND RESULTS

## D.1 IMPLEMENTATION DETAILS

We select three different noise levels $\sigma \in \{0.25, 0.5, 1.0\}$ for certification. For the parameters of DensePure , The sampling numbers when computing the certified radius are $n = 100,000$ for CIFAR-10 and $n = 10,000$ for ImageNet. We evaluate the certified robustness on 500 samples subset of CIFAR-10 testset and 100 samples subset of ImageNet validation set. we set $K = 40$ and $b = 10$ except the results in ablation study.

## D.2 BASELINES.

We select randomized smoothing based methods including PixelDP (Lecuyer et al., 2019), RS (Cohen et al., 2019), SmoothAdv (Salman et al., 2019a), Consistency (Jeong & Shin, 2020), MACER (Zhai et al., 2020), Boosting (Horváth et al., 2021) , SmoothMix (Jeong et al., 2021), Denoised (Salman et al., 2020), Lee (Lee, 2021), Carlini (Carlini et al., 2022) as our baselines. Among them, PixelDP, RS, SmoothAdv, Consistency, MACER, and SmoothMix require training a smooth classifier for a better certification performance while the others do not. Salman et al. and Lee use the off-the-shelf classifier but without using the diffusion model. The most similar one compared with us is Carlini et al., which also uses both the off-the-shelf diffusion model and classifier. The above two settings mainly refer to Carlini et al. (2022), which makes us easier to compared with their results.

## D.3 MAIN RESULTS FOR CERTIFIED ACCURACY

We compare with Carlini et al. (2022) in a more fine-grained version. We provide results of certified accuracy at different $\epsilon$ in Table A for CIFAR-10 and Table B for ImageNet. We include the accuracy difference between ours and Carlini et al. (2022) in the bracket in Tables. We can observe from the tables that the certified accuracy of our method outperforms Carlini et al. (2022) except $\epsilon = 0$ at $\sigma = 0.25, 0.5$ for CIFAR-10.

## D.4 EXPERIMENTS FOR VOTING SAMPLES

Here we provide more experiments with $\sigma \in \{0.5, 1.0\}$ and $b = 10$ for different voting samples $K$ in Figure A and Figure B. The results for CIFAR-10 is in Figure G. We can draw the same conclusion mentioned in the main context .

| Methods | Noise | Certified Accuracy at $\epsilon(\%)$ | | | | | |
|---------|-------|------|------|------|------|------|------|
| | | 0.0 | 0.5 | 1.0 | 1.5 | 2.0 | 3.0 |
| Carlini (Carlini et al., 2022) | $\sigma = 0.25$ | 77.0 | 71.0 | 0.0 | 0.0 | 0.0 | 0.0 |
| | $\sigma = 0.5$ | 74.0 | 67.0 | 54.0 | 46.0 | 0.0 | 0.0 |
| | $\sigma = 1.0$ | 59.0 | 53.0 | 49.0 | 38.0 | 29.0 | 22.0 |
| **Ours** | $\sigma = 0.25$ | **80.0(+3.0)** | **76.0(+5.0)** | 0.0(+0.0) | 0.0(+0.0) | 0.0(+0.0) | 0.0(+0.0) |
| | $\sigma = 0.5$ | 75.0(+1.0) | 72.0(+5.0) | **62.0(+8.0)** | **49.0(+3.0)** | 0.0(+0.0) | 0.0(+0.0) |
| | $\sigma = 1.0$ | 61.0(+2.0) | 57.0(+4.0) | 53.0(+4.0) | **49.0(+11.0)** | **37.0(+8.0)** | **26.0(+4.0)** |

Table B: Certified accuracy compared with Carlini et al. (2022) for ImageNet at all $\sigma$. The numbers in the bracket are the difference of certified accuracy between two methods. Our diffusion model and classifier are the same as Carlini et al. (2022).

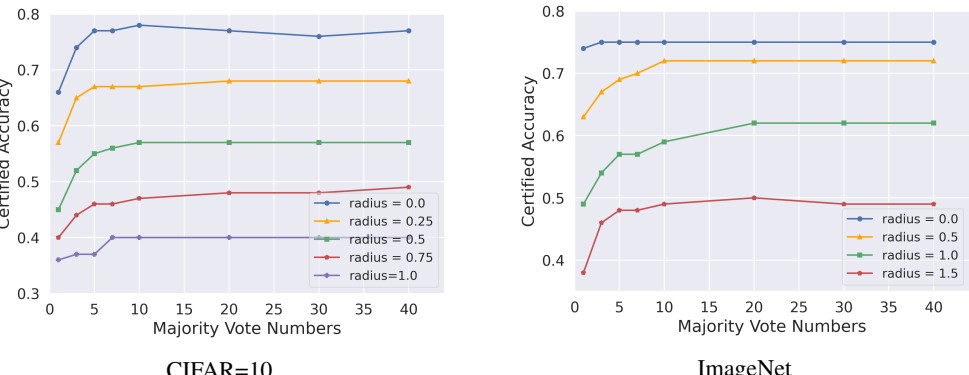

CIFAR=10                    ImageNet

Figure A: Certified accuracy among different vote numbers with different radius. Each line in the figure represents the certified accuracy among different vote numbers K with Gaussian noise $\sigma = 0.50$.

## D.5 EXPERIMENTS FOR FAST SAMPLING STEPS

We also implement additional experiments with $b \in \{1, 2, 10\}$ at $\sigma = 0.5, 1.0$. The results are shown in Figure C and Figure D. The results for CIFAR-10 are in Figure G. We draw the same conclusion as mentioned in the main context.

## D.6 EXPERIMENTS FOR DIFFERENT ARCHITECTURES

We try different model architectures of ImageNet including Wide ResNet-50-2 and ResNet 152 with $b = 2$ and $K = 10$. The results are shown in Figure F. We find that our method outperforms (Carlini et al., 2022) for all $\sigma$ among different classifiers.

## D.7 EXPERIMENTS FOR RANDOMIZED SMOOTHING WITHOUT DIFFUSION MODEL

To show the effectiveness of our diffusion model design, we remove the diffusion model from our pipeline and conduct experiments. Specifically, first, we remove the diffusion model and perform randomized smoothing only on the pretrained classifier that we used in DensePure (i.e., ViT-B/16 for CIFAR-10 and BEiT for ImageNet). The results are shown in Table C and Table D. The number in the bracket is the robust accuracy of pretrained classifier - the robust accuracy of DensePure. From the result, we conclude that without the help of diffusion models, neither ViT nor BEiT could reach high certified accuracy.

Second, we conduct additional experiments to fairly compare with randomized smoothing without diffusion models under majority vote settings. Specifically, we activate droppath in BEiT at the inference stage to support majority votes. The other settings are the same as DensePure. The results are shown in Table E. The number in the bracket is calculated by the robust accuracy of BeiT with majority votes - the robust accuracy of DensePure. We find that simply performing majority votes on the BeiT classifier will not result in higher certified robustness.

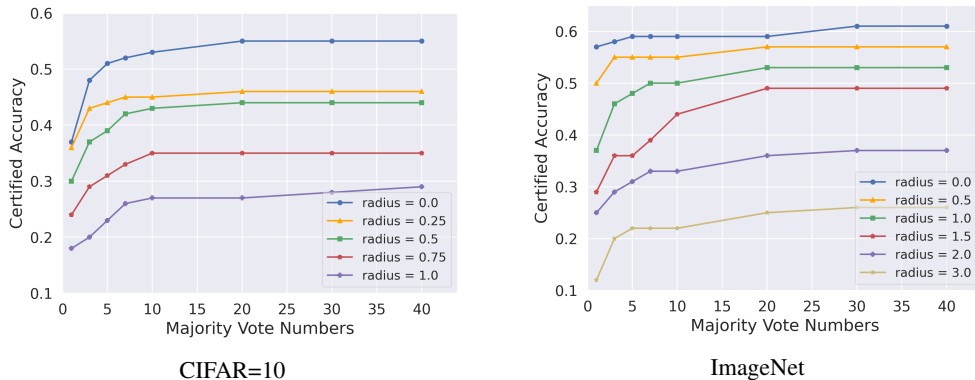

Figure B: Certified accuracy among different vote numbers with different radius. Each line in the figure represents the certified accuracy among different vote numbers K with Gaussian noise $\sigma = 1.00$.

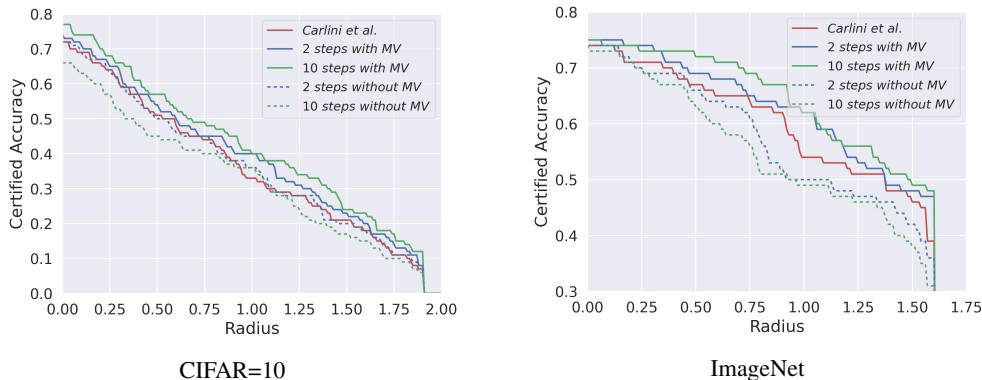

Figure C: Certified accuracy with different fast sampling steps $b$. Each line in the figure shows the certified accuracy among different $L_2$ adversarial perturbation bound with Gaussian noise $\sigma = 0.50$.

Third, to compare with randomized smoothing without diffusion model, we also evaluate certified accuracy with Gaussian augmentation-trained ViT models on CIFAR-10. The results shown in the table F prove that DensePure can still achieve higher certified accuracy than randomized smoothing on even Gaussian augmented models without diffusion models. The numbers in the bracket are the difference between the robust accuracy of Gaussian augmentation randomized smoothing and DensePure.

### D.8 EXPERIMENTS FOR K-CONSENSUS AGGREGATION

To improve the efficient of our algorithm, we try the K-consensus Aggregation, where an early stop will be triggered if the classification results of the K consecutive reversed samples are the same. Here we calculate the certified robustness for 100 subsamples of CIFAR-10 and ImageNet with 2 sampling steps, a maximum 10 majority votes and consensus threshold k=3. Results are shown in Table G and Table H. The column of "Avg MV" in the tables means the average of the actual number of majority votes required for our algorithm. For instance, if the predicted labels of the first 3 reversed samples are the same, the actual majority vote numbers will be 3. The numbers in the bracket are the difference between certified accuracy w/o K-Consensus Aggregation.

### D.9 EXPERIMENTS FOR CERTIFIED ACCURACY WITH LESS SAMPLING STEPS AND VOTE NUMBERS

We also conduct additional experiments with 2 sampling steps and 5 majority votes. The results are shown in Table I. We find that our method still achieves better results than the existing method.

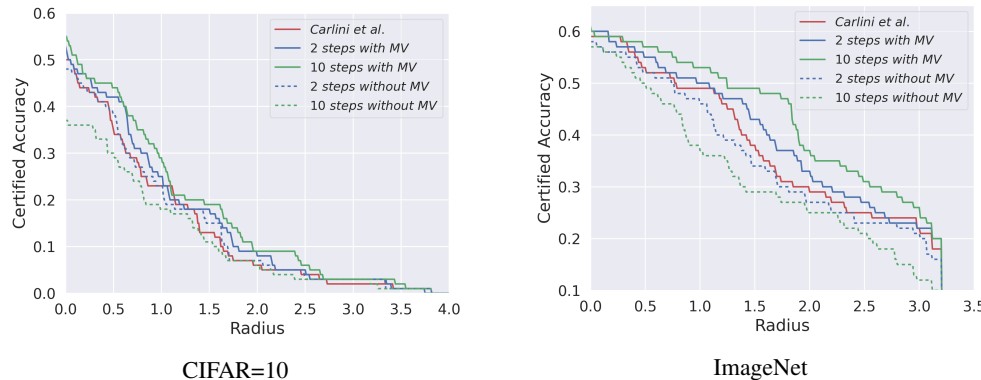

CIFAR=10 ImageNet

Figure D: Certified accuracy with different fast sampling steps $b$. Each line in the figure shows the certified accuracy among different $L_2$ adversarial perturbation bound with Gaussian noise $\sigma = 1.00$.

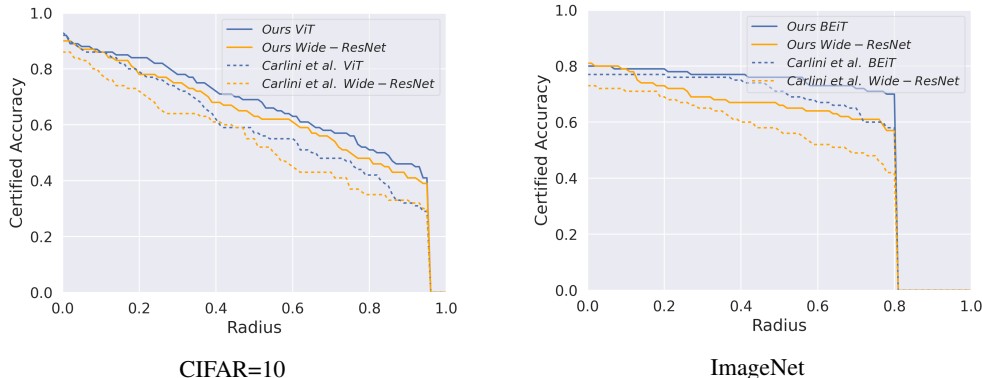

CIFAR=10 ImageNet

Figure E: Certified accuracy with different architectures. Each line in the figure shows the certified accuracy among different $L_2$ adversarial perturbation bound with Gaussian noise $\sigma = 0.25$.

### D.10 EXPERIMENTS FOR DENSEPURE 500 TEST SAMPLING NUMBER RESULTS ON IMAGENET

We increase the ImageNet test sampling number from 100 to 500 and update the experiment results in Table J and Table K. We can draw the similar conclusion.

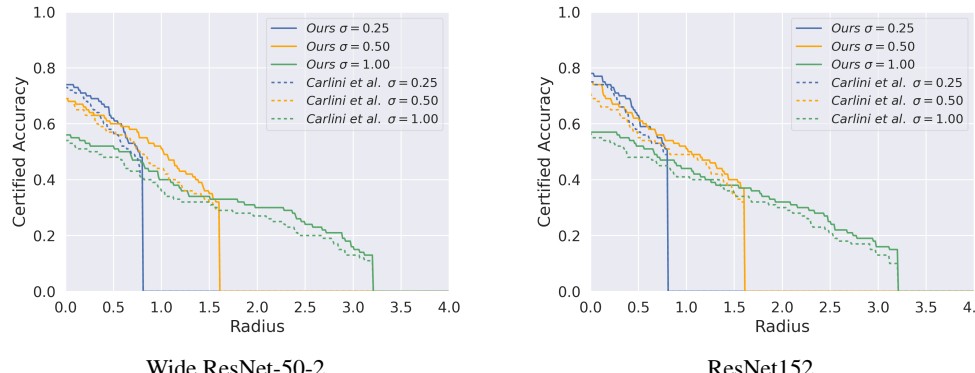

| Wide ResNet-50-2 | ResNet152 |

Figure F: Certified accuracy of ImageNet for different architectures. The lines represent the certified accuracy with different $L_2$ perturbation bound with different Gaussian noise $\sigma \in \{0.25, 0.50, 1.00\}$.

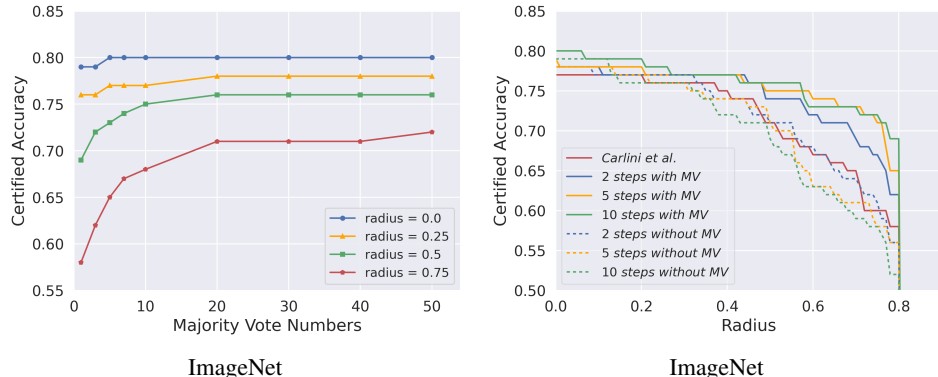

| ImageNet | ImageNet |

Figure G: Ablation study. The left image shows the certified accuracy among different vote numbers with different radius $\epsilon \in \{0.0, 0.25, 0.5, 0.75\}$. Each line in the figure represents the certified accuracy of our method among different vote numbers $K$ with Gaussian noise $\sigma = 0.25$. The right image shows the certified accuracy with different fast sampling steps $b$. Each line in the figure shows the certified accuracy among different $L_2$ adversarial perturbation bound.

| | Certified Accuracy at $\epsilon(\%)$ | | | | |
| Noise | 0.0 | 0.25 | 0.5 | 0.75 | 1.0 |
|---|---|---|---|---|---|
| $\sigma = 0.25$ | 20.8(-66.8) | 7.4(-69.2) | 1.8(-62.8) | 0.2(-50.2) | 0.0(+0.0) |
| $\sigma = 0.5$ | 11.6(-62.0) | 6.6(-58.8) | 3.8(-51.8) | 1.2(-44.8) | 0.2(-37.2) |
| $\sigma = 1.0$ | 10.6(-44.4) | 10.6(-37.4) | 9.4(-31.4) | 9.4(-23.6) | 9.4(-18.8) |

Table C: Certified accuracy of randomized smoothing on pretrained classifier ViT-B/16 at all $\sigma$ for CIFAR-10

| | Certified Accuracy at $\epsilon(\%)$ | | | | | |
| Noise | 0.0 | 0.5 | 1.0 | 1.5 | 2.0 | 3.0 |
|---|---|---|---|---|---|---|
| $\sigma = 0.25$ | 73.2(-10.8) | 55.8(-22.0) | 0.0(+0.0) | 0.0(+0.0) | 0.0(+0.0) | 0.0(+0.0) |
| $\sigma = 0.5$ | 7.8(-72.4) | 4.6(-71.0) | 3.2(-63.8) | 1.0(-53.6) | 0.0(+0.0) | 0.0(+0.0) |
| $\sigma = 1.0$ | 0.0(-67.8) | 0.0(-61.4) | 0.0(-55.6) | 0.0(-50.0) | 0.0(-42.2) | 0.0(-25.8) |

Table D: Certified accuracy of randomized smoothing on pretrained classifier BEiT at all $\sigma$ for ImageNet

| Noise | Certified Accuracy at $\epsilon(\%)$ | | | | | |
|---|---|---|---|---|---|---|
| | 0.0 | 0.5 | 1.0 | 1.5 | 2.0 | 3.0 |
| $\sigma = 0.25$ | 73.8(-10.2) | 58.0(-19.8) | 0.0(+0.0) | 0.0(+0.0) | 0.0(+0.0) | 0.0(+0.0) |
| $\sigma = 0.5$ | 9.0(-71.2) | 7.0(-68.6) | 4.0(-63.0) | 2.0(-52.6) | 0.0(+0.0) | 0.0(+0.0) |
| $\sigma = 1.0$ | 0.0(-67.8) | 0.0(-61.4) | 0.0(-55.6) | 0.0(-50.0) | 0.0(-42.2) | 0.0(-25.8) |

Table E: Certified accuracy of randomized smoothing on droppatch activated BEiT with 10 majority votes at all $\sigma$ for ImageNet

| Noise | Certified Accuracy at $\epsilon(\%)$ | | | | |
|---|---|---|---|---|---|
| | 0.0 | 0.25 | 0.5 | 0.75 | 1.0 |
| $\sigma = 0.25$ | 88.2(+0.6) | 71.4(-5.2) | 53.2(-11.4) | 35.2(-15.2) | 0.0(+0.0) |
| $\sigma = 0.5$ | 69.8(-3.8) | 60.0(-5.4) | 48.4(-7.2) | 37.2(-8.8) | 27.2(-10.2) |
| $\sigma = 1.0$ | 49.0(-6.0) | 41.8(-6.0) | 34.0(-6.8) | 27.0(-6.0) | 22.0(-6.2) |

Table F: Certified accuracy of randomized smoothing on Gaussian augmentation-trained ViT at all $\sigma$ on CIFAR-10

| Noise | Certified Accuracy at $\epsilon(\%)$ | | | | | Avg MV |
|---|---|---|---|---|---|---|
| | 0.0 | 0.25 | 0.5 | 0.75 | 1.0 | |
| $\sigma = 0.25$ | 92(+0.0) | 77(+0.0) | 60(+0.0) | 48(-1.0) | 0(+0.0) | 3.84 |
| $\sigma = 0.5$ | 74(+0.0) | 65(+0.0) | 53(-1.0) | 45(+0.0) | 40(+0.0) | 4.43 |
| $\sigma = 1.0$ | 53(+0.0) | 46(+0.0) | 42(+0.0) | 31(+0.0) | 25(+0.0) | 5.49 |

Table G: Certified accuracy and average majority votes with 2 sample steps and $k = 3$ consensus threshold at all $\sigma$ for CIFAR-10.

| Noise | Certified Accuracy at $\epsilon(\%)$ | | | | | | Avg MV |
|---|---|---|---|---|---|---|---|
| | 0.0 | 0.5 | 1.0 | 1.5 | 2.0 | 3.0 | |
| $\sigma = 0.25$ | 78(+0.0) | 74(+0.0) | 0(+0.0) | 0(+0.0) | 0(+0.0) | 0(+0.0) | 3.34 |
| $\sigma = 0.5$ | 75(+0.0) | 69(+0.0) | 61(+0.0) | 47(+0.0) | 0(+0.0) | 0(+0.0) | 3.89 |
| $\sigma = 1.0$ | 60(+0.0) | 54(+0.0) | 50(+0.0) | 41(+0.0) | 32(+0.0) | 23(+0.0) | 5.23 |

Table H: Certified accuracy and average majority votes with 2 sample steps and $k = 3$ consensus threshold at all $\sigma$ for ImageNet.

| Noise | Certified Accuracy at $\epsilon(\%)$ | | | | | | | | | | |
|---|---|---|---|---|---|---|---|---|---|---|---|
| | CIFAR-10 | | | | | ImageNet | | | | | |
| | 0.0 | 0.25 | 0.5 | 0.75 | 1.0 | 0.0 | 0.5 | 1.0 | 1.5 | 2.0 | 3.0 |
| $\sigma = 0.25$ | 87.6 | 74.8 | 59.2 | 44.6 | 0.0 | 78 | 74 | 0 | 0 | 0 | 0 |
| $\sigma = 0.50$ | 73.2 | 62.6 | 52.6 | 41.8 | 34.0 | 75 | 69 | 58 | 47 | 0 | 0 |
| $\sigma = 1.00$ | 53.4 | 44.0 | 35.8 | 30.2 | 24.4 | 60 | 54 | 49 | 39 | 30 | 22 |

Table I: Certified accuracy with 2 sampling steps and 5 vote numbers at all $\sigma$ for both CIFAR-10 and ImageNet

| | | Certified Accuracy at $\epsilon$(%) | | | | | | | | |
|---|---|---|---|---|---|---|---|---|---|---|
| | | CIFAR-10 | | | | | ImageNet | | | |
| Method | Off-the-shelf | 0.25 | 0.5 | 0.75 | 1.0 | 0.5 | 1.0 | 1.5 | 2.0 | 3.0 |
| PixelDP (Lecuyer et al., 2019) | ✗ | $^{(71.0)}$22.0 | $^{(44.0)}$2.0 | - | - | $^{(33.0)}$16.0 | - | - | - | - |
| RS (Cohen et al., 2019) | ✗ | $^{(75.0)}$61.0 | $^{(75.0)}$43.0 | $^{(65.0)}$32.0 | $^{(65.0)}$23.0 | $^{(67.0)}$49.0 | $^{(57.0)}$37.0 | $^{(57.0)}$29.0 | $^{(44.0)}$19.0 | $^{(44.0)}$12.0 |
| SmoothAdv (Salman et al., 2019a) | ✗ | $^{(82.0)}$68.0 | $^{(76.0)}$54.0 | $^{(68.0)}$41.0 | $^{(64.0)}$32.0 | $^{(63.0)}$54.0 | $^{(56.0)}$42.0 | $^{(56.0)}$34.0 | $^{(41.0)}$26.0 | $^{(41.0)}$18.0 |
| Consistency (Jeong & Shin, 2020) | ✗ | $^{(77.8)}$68.8 | $^{(75.8)}$58.1 | $^{(72.9)}$48.5 | $^{(52.3)}$37.8 | $^{(55.0)}$50.0 | $^{(55.0)}$44.0 | $^{(55.0)}$34.0 | $^{(41.0)}$24.0 | $^{(41.0)}$17.0 |
| MACER (Zhai et al., 2020) | ✗ | $^{(81.0)}$71.0 | $^{(81.0)}$59.0 | $^{(66.0)}$46.0 | $^{(66.0)}$38.0 | $^{(68.0)}$57.0 | $^{(64.0)}$43.0 | $^{(64.0)}$31.0 | $^{(48.0)}$25.0 | $^{(48.0)}$14.0 |
| Boosting (Horváth et al., 2021) | ✗ | $^{(83.4)}$70.6 | $^{(76.8)}$60.4 | $^{(71.6)}$**52.4** | $^{(73.0)}$**38.8** | $^{(65.6)}$57.0 | $^{(57.0)}$44.6 | $^{(57.0)}$38.4 | $^{(44.6)}$28.6 | $^{(38.6)}$21.2 |
| SmoothMix (Jeong et al., 2021) | ✓ | $^{(77.1)}$67.9 | $^{(77.1)}$57.9 | $^{(74.2)}$47.7 | $^{(61.8)}$37.2 | $^{(55.0)}$50.0 | $^{(55.0)}$43.0 | $^{(55.0)}$38.0 | $^{(40.0)}$26.0 | $^{(40.0)}$17.0 |
| Denoised (Salman et al., 2020) | ✓ | $^{(72.0)}$56.0 | $^{(62.0)}$41.0 | $^{(62.0)}$28.0 | $^{(44.0)}$19.0 | $^{(60.0)}$33.0 | $^{(38.0)}$14.0 | $^{(38.0)}$6.0 | - | - |
| Lee (Lee, 2021) | ✓ | 60.0 | 42.0 | 28.0 | 19.0 | 41.0 | 24.0 | 11.0 | - | - |
| Carlini (Carlini et al., 2022) | ✓ | $^{(88.0)}$73.8 | $^{(88.0)}$56.2 | $^{(88.0)}$41.6 | $^{(74.2)}$31.0 | $^{(82.0)}$74.0 | $^{(77.2.0)}$59.8 | $^{(77.2)}$47.0 | $^{(64.6)}$31.0 | $^{(64.6)}$19.0 |
| **Ours** | ✓ | $^{(87.6)}$**76.6** | $^{(87.6)}$**64.6** | $^{(87.6)}$**50.4** | $^{(73.6)}$37.4 | $^{(84.0)}$**77.8** | $^{(80.2)}$**67.0** | $^{(80.2)}$**54.6** | $^{(67.8)}$**42.2** | $^{(67.8)}$**25.8** |

Table J: Certified accuracy compared with existing works. The certified accuracy at $\epsilon = 0$ for each model is in the parentheses. The certified accuracy for each cell is from the respective papers except Carlini et al. (2022). Our diffusion model and classifier are the same as Carlini et al. (2022), where the off-the-shelf classifier uses ViT-based architectures trained on a large dataset (ImageNet-22k).

| | | Certified Accuracy at $\epsilon$(%) | | | | | |
|---|---|---|---|---|---|---|---|
| Methods | Noise | 0.0 | 0.5 | 1.0 | 1.5 | 2.0 | 3.0 |
| | $\sigma = 0.25$ | 82.0 | 74.0 | 0.0 | 0.0 | 0.0 | 0.0 |
| Carlini (Carlini et al., 2022) | $\sigma = 0.5$ | 77.2 | 71.8 | 59.8 | 47.0 | 0.0 | 0.0 |
| | $\sigma = 1.0$ | 64.6 | 57.8 | 49.2 | 40.6 | 31.0 | 19.0 |
| | $\sigma = 0.25$ | **84.0(+2.0)** | **77.8(+3.8)** | 0.0(+0.0) | 0.0(+0.0) | 0.0(+0.0) | 0.0(+0.0) |
| **Ours** | $\sigma = 0.5$ | 80.2(+3.0) | 75.6(+3.8) | **67.0(+7.2)** | **54.6(+7.6)** | 0.0(+0.0) | 0.0(+0.0) |
| | $\sigma = 1.0$ | 67.8(+3.2) | 61.4(+3.6) | 55.6(+6.4) | 50.0(+9.4) | **42.2(+11.2)** | **25.8(+6.8)** |

Table K: Certified accuracy compared with Carlini et al. (2022) for ImageNet at all $\sigma$. The numbers in the bracket are the difference of certified accuracy between two methods. Our diffusion model and classifier are the same as Carlini et al. (2022).

