# OpenReview forum: "DensePure: Understanding Diffusion Models for Adversarial Robustness"
_ICLR.cc/2023/Conference — ICLR 2023 poster_

### Official Review · Reviewer_pQBA · 2022-10-21

**Confidence:** 2
**Correctness:** 3
**Technical Novelty And Significance:** 3
**Empirical Novelty And Significance:** 3
**Recommendation:** 8

**Clarity, Quality, Novelty And Reproducibility:**

Sections 1 and 2 are well written and could be understand by a reader unfamiliar with adversarial attacks/robustness (which is my case).
I made the educated guest for section 3 and 4.
In section 5, experimentation are well described and reproducible.

**Strength And Weaknesses:**

Strength:
- simplicity of the framework
- use of off the shelves diffusion model and classifiers

**Summary Of The Paper:**

This paper proposed to increased robustness to adversarial attacks of an off the shelf classifier by using a (of the shelf) diffusion model as a data augmentation preprocessing step.



**Summary Of The Review:**

The framework is simple and clever.
I could not verify the theoretical analysis part as it is out of my skills.

---

> ### Author Response · Authors · 2022-11-19
> **Response to Reviewer pQBA**
>
> Thank you for the review.
>
> > **Summary Of The Review:**The framework is simple and clever. I could not verify the theoretical analysis part as it is out of my skills.
>
> Response: We thanks the reviewer agrees that our framework is simple and clever. We will add more explanation for our theoretical analysis to make it more readable to general audiences.

---

### Official Review · Reviewer_HvUT · 2022-10-24

**Confidence:** 4
**Correctness:** 2
**Technical Novelty And Significance:** 3
**Empirical Novelty And Significance:** 4
**Recommendation:** 8

**Clarity, Quality, Novelty And Reproducibility:**

Clarity score 7/10: In general, this paper is well-written. However, it still has typos. For example, "the robust region for data region with ground-truth label under $\mathbb{P} (\cdot; t)$" should be "the robust region for data region with ground-truth label under $\mathcal{P} (\cdot; t)$".

Quality score 6/10: The theoretical results as well as the proposed DensePure are technically sound. However, the first claimed contribution seems not well supported. (See Weakness (W2) for more details. If I misunderstood this paper, the authors please point it out directly.)

Novelty score 7/10: To the best of my knowledge, this paper theoretically analyzes why and how diffusion model performs well in purification by proposing three interesting and novel theorems.

Reproductivity score: 6/10: Although the  code is not shared, I think the details in the supplementary materials are sufficient for reproduction.


**Strength And Weaknesses:**

Strengths:

S1) Theoretical strength: This paper theoretically analyzes why and how diffusion-based purification model can enhance the adversarial robustness of a given classifier for the first time. The robust region of a sample (under a deterministic inverse purification process and a base classifier)  is characterized as the union of several convex sub-robust regions (indexed by samples with the same ground truth label). The newly characterized robust region may provide with a larger robust radius than other methods, which indicates better robustness.

S2) Modelling/algorithmic strength:  The proposed DensePure performs consistently better than existing methods on ImageNet, with 7% improvement on average.

Weaknesses:

W1) Time complexity of DensePure: As it requires repeating the reverse process multiple times, DensePure is really time-consuming and this drawback may prevent it from further applications to large datasets.

W2) Claimed but seemingly not well-supported contributions:  The authors claimed their first contributions as "We prove that under constrained data density property, an adversarial example can be recovered back to the original clean sample with high probability via the reverse process of a diffusion model". However, it seems not well-supported. The reason is as follows:

Let $x_0$ be the clean sample and $x_a$ be its adversarially perturbed sample.
Theorem 3.1 characterizes the distribution of inversed variable $\hat{x}_0$ from a scaled adversary $x\_{a,t}=\sqrt{\alpha}\_t x\_a$ as
$$ \mathbb{P}(\hat{x}_0=x|\hat{x}\_t=x\_{a,t}) \propto p(x)\cdot\frac{1}{\sqrt{(2\pi\sigma^2_t)^n}}\textnormal{exp}\big(\frac{-\|\|x-x_a\|\|_2^2}{2\sigma^2_t}\big)$$ Letting $x=x_0$, we obtain
$$ \mathbb{P}(\hat{x}_0=x_0|\hat{x}\_t=x\_{a,t}) \propto p(x_0)\cdot\frac{1}{\sqrt{(2\pi\sigma^2_t)^n}}\textnormal{exp}\big(\frac{-\|\|x_0-x_a\|\|_2^2}{2\sigma^2_t}\big)$$

which seems unable to directly imply "$\mathbb{P}(\hat{x}_0=x_0|\hat{x}\_t=x\_{a,t})$ is high" or "$\mathbb{P}(\|\|\hat{x}_0-x_0\|\|\le \delta |\hat{x}\_t=x\_{a,t})$ is high with a small $\delta$".

So, I failed to understand why "under constrained data density property, an adversarial example can be recovered back to the original clean sample with high probability". What is the "constrained data density property"? Can the authors provide upper bounds on $\|\|\hat{x}_0-x_0\|\|$ to verify "an adversarial example can be recovered back to the original clean sample with high probability"? Did I miss something?

----------------After rebuttal------------
In the rebuttal, both of my main concerns about "Weakness 1: Time complexity of DensePure" and "Weakness 2: Claimed but seemingly not well-supported contributions" have been well explained. Therefore, I decided to change my score from 6 to 8.

**Summary Of The Paper:**

This paper studies the conditions under which diffusion model can work well for purification of adversarially perturbed samples. A simple diffusion-based purification method named DensePure is proposed by using majority vote  and achieves higher certified accuracy in comparison with other method on CIFAR-10 and ImageNet.

**Summary Of The Review:**

In general, this paper provides novel theoretical results and an effective algorithm.  I suggest "marginally above the acceptance threshold" mainly due to "Weakness (W2) Claimed but seemingly not well-supported contributions". See Weakness (W2) for more details.

---

> ### Author Response · Authors · 2022-11-19
> **Response to Reviewer HvUT (Part 2)**
>
> >**Weakness2:**
> >Claimed but seemingly not well-supported contributions: The authors claimed their first contributions as "We prove that under constrained data density property, an adversarial example can be recovered back to the original clean sample with high probability via the reverse process of a diffusion model". However, it seems not well-supported. The reason is as follows:
> >Let $x_0$ be the clean sample and $x_a$ be its adversarially perturbed sample. Theorem 3.1 characterizes the distribution of inversed variable $\hat x_0$ from a scaled adversary $x_{a,t} = \sqrt{α}_tx_a$ as ......
> >So, I failed to understand why "under constrained data density property, an adversarial example can be recovered back to the original clean sample with high probability". What is the "constrained data density property"? Can the authors provide upper bounds on $||\hat{x}_0−x_0||$ to verify "an adversarial example can be recovered back to the original clean sample with high probability"? Did I miss something?
>
> Response: $\mathbb{P}\left(\hat x_0 =x| {\hat x_t = x_{a,t}}\right)$ is high if both $p(x_0)$ is high and $||x_0 - x_a||_2$ is small. When we are saying, "We prove that under constrained data density property, an adversarial example can be recovered back to the original clean sample with high probability via the reverse process of a diffusion model", we are actually referring to Theorem 3.3 where we characterized the robust region for the data manifold corresponding to the label of the clean sample. Our claim is that when the constrained data property holds, such robust regions are large so the adversarial examples will be in them with high probability and can be recovered back to the original clean sample with high probability.
>
> "Constrained data density property" means that points with the ground-truth label should have relatively high density and are separated away from points with different labels. With this property, Theorem 3.3 implies that the robust region/ robust radius is large for the data manifold corresponding to the label of the clean sample. Since the adversarial attack is typically small, with a large robust region/ robust radius, it will highly likely locate within the robust region/ robust radius, which means we can recover the adversarial sample from the cleaning sample with a high probability.
>
> This is a reasonable assumption because on the one hand, if the points with the ground-truth label have a relatively low density in the original data distribution, we cannot expect the diffusion model to generate them with high probability and generate high quality images. Thus we cannot expect the diffusion model to purify the corresponding adversarial samples well. On the other hand, if the points with the ground-truth label are not separated from other points, it is hard to distinguish which is the true sample of the adversarial sample.
>
> Intuitively, every training sample with the ground-truth label in the diffusion model is with high density and are separated from other points. Because the diffusion model will recover a distribution similar to the training data distribution. In the training data distribution, each training sample will correspond to an impulse, and thus a high density in the recovered distribution and separated.

---

> > ### Comment · Reviewer_HvUT · 2022-11-23
> > **Response to the authors' feedback**
> >
> > Many thanks for the authors' feedback. Both of my main concerns about "Weakness 1: Time complexity of DensePure" and "Weakness 2: Claimed but seemingly not well-supported contributions" have been well explained. Therefore, I decided to change my score from 6 to 8.

---

> ### Author Response · Authors · 2022-11-19
> **Response to Reviewer HvUT (Part 1-2)**
>
> Furthermore, there is a large potential to improve the time efficiency of DensePure. For instance,  we can use K-Consensus Aggregation proposed in the paper [1] to improve the efficiency of the majority voting of DensePure. In this algorithm, if the classification results of the K consecutive reversed samples are the same, an early stop will be triggered.
>
> We conduct additional experiments by using K-consensus aggregation. We calculate certified robustness for 100 subsamples of CIFAR-10 and ImageNet with 2 sampling steps, a maximum 10 majority votes and consensus threshold k=3. The results are shown as follows:
>
>
>
>
>  CIFAR-10
>
>  | Certified Accuracy at $\epsilon$(%) | 0.00 | 0.25 | 0.50 | 0.75 | 1.00 | Avg majority votes |
>  | ------------- | ---- | ---- | ---- | ---- | ---- | ------------------ |
>  | $\sigma=0.25$          | 92(+0.0)   | 77(+0.0)   | 60(+0.0)   | 48(-1.0)   | 0(+0.0)    | 3.84               |
>  | $\sigma=0.50$          | 74(+0.0)   | 65(+0.0)   | 53(-1.0)   | 45(+0.0)   | 40(+0.0)   | 4.43               |
>  | $\sigma=1.00$          | 53(+0.0)   | 46(+0.0)   | 42(+0.0)   | 31(+0.0)   | 25(+0.0)   | 5.49               |
>
>
>
>  ImageNet
>
>  | Certified Accuracy at $\epsilon$(%) | 0.00 | 0.25 | 0.50 | 0.75 | 1.00 | 3.00 | Avg majority votes |
>  | ------------- | ---- | ---- | ---- | ---- | ---- | ---- | ------------------ |
>  | $\sigma=0.25$          | 78(+0.0)   | 74(+0.0)   | 0(+0.0)    | 0(+0.0)    | 0(+0.0)    | 0(+0.0)    | 3.34               |
>  | $\sigma=0.50$          | 75(+0.0)   | 69(+0.0)   | 61(+0.0)   | 47(+0.0)   | 0(+0.0)    | 0(+0.0)    | 3.89               |
>  | $\sigma=1.00$          | 60(+0.0)   | 54(+0.0)   | 50(+0.0)   | 41(+0.0)   | 32(+0.0)   | 23(+0.0)   | 5.23               |
>
>
> The column of avg majority votes means the average of the actual number of majority votes required for our algorithm. For instance,  if the predicted labels of the first 3 reversed samples are the same, the actual majority vote numbers will be 3. The numbers in the bracket are the difference between certified accuracy w/o K-Consensus Aggregation.
>
>
> We observe that comparing with 2 sampling steps 10 majority votes without K-Consensus Aggregation, using K-Consensus Aggregation can make the majority vote process 45%~60% faster with almost no performance degradation with respect to the certified accuracy.
>
>
>
> Moreover, the time efficiency of DensePure can be further improved by accelerating randomized smoothing algorithms or even the sampling speed of diffusion models. For instance:
>
>   1. In paper [1], the adaptive sampling method for randomized smoothing shown in Section 6 can also be used in our DensePure pipeline to make faster-randomized smoothing.
>
>   2. To improve the sampling speed of diffusion models, we can use the distilled diffusion model in paper [2], which can generate high-quality images in only 4 steps.
>
> We will add related discussions in our revision.
>
> [1] Miklos Z Horvath, Mark Niklas Muller, Marc Fischer, and Martin Vechev. Boosting randomized smoothing with variance reduced classifiers. arXiv preprint arXiv:2106.06946, 2021.
>
> [2]Tim Salimans, Jonathan Ho. Progressive Distillation for Fast Sampling of Diffusion Models. ICLR 2022

---

> ### Author Response · Authors · 2022-11-19
> **Response to Reviewer HvUT (Part 1-1)**
>
> Thank you for the review.
>
> >**Weakness1:**
> >Time complexity of DensePure: As it requires repeating the reverse process multiple times, DensePure is really time-consuming, and this drawback may prevent it from further applications to large datasets.
>
> Response:
> We hope to highlight that the main goal of this paper is to provide a theoretical understanding of why and how the adversarial purification by diffusion models contributes to adversarial robustness, which has not been explored.
>
> To the best of our knowledge, (1) we are the first to prove that under constrained data density property (i.e., points with the ground-truth label should have relatively high density and are far away from points with different labels), an adversarial example can be recovered back to the original clean sample with a high probability via the reverse process of a diffusion model; (2) we are the first to characterize the robust region of diffusion models against adversarial attacks and show that the robust region of a given sample under the diffusion model’s reverse process has the potential to provide a larger robust region.
>
> The certification algorithm, DensePure, mainly aims to empirically verify our theoretical understanding and needs to follow our theoretical understanding. Our empirical results confirm that DensePure can achieve higher certified robustness than existing methods.
>
>
>
> Within this context, how to further improve the time efficiency of DensePure is an open problem. In Figure 4 and Appendix D.5 (Figure C,D,G), we reduce the reverse sampling steps and majority vote numbers to improve efficiency. We show that with even 2 sampling steps and 10 majority votes (which can largely reduce the repeating times), we can still achieve non-trivial certified robustness. Additionally, we also conduct additional experiments with 2 sampling steps and 5 majority votes. The results are shown as follows. We  find that our method still achieves better results than the existing method.
>
> CIFAR-10
>
> | Certified Accuracy at $\epsilon$(%) | 0.00 | 0.25 | 0.50 | 0.75 | 1.00 |
> | ------------- | ---- | ---- | ---- | ---- | ---- |
> | $\sigma=0.25$          | 87.6 | 74.8 | 59.2 | 44.6 | 0.0    |
> | $\sigma=0.50$          | 73.2 | 62.6 | 52.6 | 41.8 | 34.0 |
> | $\sigma=1.00$          | 53.4 | 44.0 | 35.8 | 30.2 | 24.4 |
>
> ImageNet
>
> | Certified Accuracy at $\epsilon$(%) | 0.0  | 0.5  | 1.0  | 1.5  | 2.0  | 3.0  |
> | ------------- | ---- | ---- | ---- | ---- | ---- | ---- |
> | $\sigma=0.25$          | 78   | 74   | 0    | 0    | 0    | 0    |
> | $\sigma=0.50$          | 75   | 69   | 58   | 47   | 0    | 0    |
> | $\sigma=1.00$          | 60   | 54   | 49   | 39   | 30   | 22   |

---

### Official Review · Reviewer_MqVZ · 2022-10-25

**Confidence:** 4
**Correctness:** 3
**Technical Novelty And Significance:** 2
**Empirical Novelty And Significance:** 2
**Recommendation:** 6

**Clarity, Quality, Novelty And Reproducibility:**

The paper is clear and easy to follow. The idea to include denoising step with diffusion model to improve classification performance is incremental. And more experiments are needed in Table 2 (see above for details)

**Strength And Weaknesses:**

Quality/Clarity: the paper is well written and the techniques presented are easy to follow. Its motivation is clear to use diffusion model to denoise input to improve classification accuracy. On a technical level, I do not see much new contribution, where most equations are from diffusion model. For the BEiT in Table 2, do we use the same BEiT model for evaluation while comparing to Carlini 2022? We do not know its gain from the model itself or from the label voting. Also it is better to add experimental comparision w/o diffusion model.

Originality/significance: the idea is interesting, which uses diffusion model to improve data quality to improve classification performance. Another contribution is the label voting from multiple samples. However, both diffusion model and pretrained classifier are known, which makes DensePure an incremental approach. In addition, the diffusion model is only for Gaussian noise, which limit the application of this approach.

**Summary Of The Paper:**

This paper proposes a new method, DensePure, designed to improve the certified robustness of a pretrained model (i.e. classifier). Specifically, DensePure uses the diffusion model to denoise the adversarial input to get multiple reversed samples, which are then passed through the off-the-shelf classifier, followed by majority voting of inferred labels to make the final prediction. The extensive experiments demonstrate the effectiveness of DensePure by evaluating its certified robustness given a standard model and show it is consistently better than existing methods on ImageNet.

**Summary Of The Review:**

DensePure with the diffusion step can improve the robustness of prediction in the late stage. Overall, it is a good paper, I am ok if it is accepted.

---

> ### Author Response · Authors · 2022-11-19
> **Response to Reviewer MqVZ (Part 3)**
>
> > Weakness on significance: " However, both diffusion model and pretrained classifier are known, which makes DensePure an incremental approach."
>
> Response: Thanks for pointing that out. Although the diffusion model and pretrained classifier are known, understanding why and how the diffusion model purifies adversarial attacks to improve the adversarial robustness is non-trivial. To the best of our knowledge, there has been no such theoretical understanding before.
>
> Compared with a significant concurrent work [5] that combines a pretrained diffusion model and classifier to improve adversarial robustness,  our design based on our theoretical analysis can achieve higher certified robustness. It further shows the importance of correctly combining pretrained diffusion model and classifier, and our theoretical analysis is critical.
>
> Moreover, we think using the pretrained diffusion model and classifier without any finetuning or retraining is indeed excellent. In this way, the users or service providers do not need to retrain the models, and models can provide a "free" lunch for adversarial robustness.
>
>
> Furthermore, our approach can either utilize off-the-shelf models to skip the model training process or train our own models. In fact, we can extend our framework to train the classifier with the reversed examples from diffusion models. We believe it can further improve the results. However, due to the time and resource limitations of the rebuttal process,  we will add the results later.
>
> Last, we had the ablation studies on different pretrained classifiers shown in Section 5.2 and Appendix D.6(Table 2 and Figure E,F.). The results show DensePure can enhance certified robustness with any given classifiers trained on the original data distribution.
>
> > Weakness: "In addition, the diffusion model is only for Gaussian noise, which limit the application of this approach."
>
> Response:
> We clarify that many diffusion models are not based on Gaussian noise, such as [1-4], and their performances are starting to match Gaussian noise-based diffusion models [3,4].
>
> Our theoretical results and the framework can be generalized to any noise distribution. However, under other noise distributions, the robust regions and subregions are much harder to characterize since they lose convexity in general. We choose to use diffusion models based on Gaussian noise to (1) get a more tractable characterization of the robustness of DensePure and (2) make better comparisons with previous works, which are mostly based on Gaussian noise. We leave the extension of our method to non-Gaussian noise-based diffusion models as future work.
>
> [1] Rissanen et al., Generative Modelling With Inverse Heat Dissipation, https://arxiv.org/abs/2206.13397.
>
> [2] Bansal et al., Cold Diffusion: Inverting Arbitrary Image Transforms Without Noise, https://arxiv.org/abs/2208.09392.
>
> [3] Hoogeboom et al., Blurring Diffusion Models, https://arxiv.org/abs/2209.05557.
>
> [4] Daras et al., Soft Diffusion: Score Matching for General Corruptions, https://arxiv.org/abs/2209.05442
>
> [5] Carlini, Nicholas, Florian Tramer, and J. Zico Kolter. "(Certified!!) Adversarial Robustness for Free!." arXiv preprint arXiv:2206.10550 (2022).

---

> ### Author Response · Authors · 2022-11-19
> **Response to Reviewer MqVZ (Part 2)**
>
> > Question: "For the BEiT in Table 2, do we use the same BEiT model for evaluation while comparing to Carlini 2022?"
>
> Response: Yes, we strictly follow all settings in [Carlini 2022] for a fair comparison. Our paper and [Carlini 2022] all directly load the BEiT pre-trained model from timm( (https://github.com/rwightman/pytorch-image-models/blob/main/results/results-imagenet.csv), which can reach 88.6% top-1 accuracy on ImageNet.
>
> > Comment: "We do not know its gain from the model itself or from the label voting. Also it is better to add experimental comparision w/o diffusion model."
>
> Response: Thanks for your suggestion. We directly remove the diffusion model from our pipeline and conduct additional experiments. First, we remove the diffusion model and perform randomized smoothing only on the pretrained classifier we used in DensePure (i.e.,  ViT-B/16 for CIFAR-10 and BEiT for ImageNet).
> The results are shown in the following. The number in the bracket is calculated by the robust accuracy of pretrained classifier - the robust accuracy of DensePure. We can conclude from the table  that without the help of diffusion models, neither ViT nor BEiT could reach high certified accuracy.
>
>
>
> CIFAR-10
>
> | Certified Accuracy at $\epsilon$(%) | 0.00 | 0.25 | 0.50 | 0.75 | 1.00 |
> | ------------- | ---- | ---- | ---- | ---- | ---- |
> | $\sigma=0.25$          | 20.8(-66.8)| 7.4(-69.2)  | 1.8(-62.8)  | 0.2(-50.2)  | 0.0(+0.0)  |
> | $\sigma=0.50$          | 11.6(-62.0) | 6.6(-58.8)  | 3.8(-51.8)  | 1.2(-44.8)  | 0.2(-37.2)  |
> | $\sigma=1.00$          | 10.6(-44.4) | 10.6(-37.4) | 9.4(-31.4)  | 9.4(-23.6)  | 9.4(-18.8)  |
>
> ImageNet
>
> | Certified Accuracy at $\epsilon$(%) | 0.0  | 0.5  | 1.0  | 1.5  | 2.0  | 3.0  |
> | ------------- | ---- | ---- | ---- | ---- | ---- | ---- |
> | $\sigma=0.25$          | 73.2(-10.8) | 55.8(-22.0) | 0.0(+0.0)  | 0.0(+0.0)  | 0.0(+0.0)  | 0.0(+0.0)  |
> | $\sigma=0.50$          | 7.8(-72.4)  | 4.6(-71.0)  | 3.2(-63.8)  | 1.0(-53.6)  | 0.0(+0.0)  | 0.0(+0.0)  |
> | $\sigma=1.00$          | 0.0(-67.8)  | 0.0(-61.4)  | 0.0(-55.6)  | 0.0(-50.0)  | 0.0(-42.2)  | 0.0(-25.8)  |
>
>
>
>
> Second, we conduct additional experiments to fairly compare with randomized smoothing without diffusion models under majority vote settings. Specifically, we activate droppath in BEiT at the inference stage to support majority votes. The other settings are the same as DensePure. The results are shown in the following table. The number in the bracket is calculated by the robust accuracy of BeiT with majority votes - the robust accuracy of DensePure.
> We show that simply performing majority votes on the BeiT classifier will not result in higher certified robustness.
>
>
> | Certified Accuracy at $\epsilon$(%) | 0.0  | 0.5  | 1.0  | 1.5  | 2.0  | 3.0  |
> | ------------- | ---- | ---- | ---- | ---- | ---- | ---- |
> | $\sigma=0.25$          | 73.8(-10.2) | 58.0(-19.8) | 0.0(+0.0)  | 0.0(+0.0)  | 0.0(+0.0)  | 0.0(+0.0)  |
> | $\sigma=0.50$          | 9.0(-71.2)  | 7.0(-68.6)  | 4.0(-63.0)  | 2.0(-52.6)  | 0.0(+0.0)  | 0.0(+0.0)  |
> | $\sigma=1.00$          | 0.0(-67.8)  | 0.0(-61.4)  | 0.0(-55.6)  | 0.0(-50.0)  | 0.0(-42.2)  | 0.0(-25.8)  |
>
>
>
>
> Third, to compare with randomized smoothing without diffusion model, we also evaluate certified accuracy with Gaussian augmentation-trained ViT models on CIFAR-10. The results shown in the table prove that DensePure can still achieve higher certified accuracy than randomized smoothing on even Gaussian augmented models without diffusion models. The numbers in the bracket are the difference between the robust accuracy of Gaussian augmentation randomized smoothing and DensePure.
>
> CIFAR-10
>
> | Certified Accuracy at $\epsilon$(%) | 0.00 | 0.25 | 0.50 | 0.75 | 1.00 |
> | ------------- | ---- | ---- | ---- | ---- | ---- |
> | $\sigma=0.25$          | 88.2(+0.6) | 71.4(-5.2)  | 53.2(-11.4)  | 35.2(-15.2)  | 0.0(+0.0)  |
> | $\sigma=0.50$          | 69.8(-3.8) | 60.0(-5.4)  | 48.4(-7.2)  | 37.2(-8.8)  | 27.2(-10.2)  |
> | $\sigma=1.00$          | 49.0(-6.0) | 41.8(-6.0) | 34.0(-6.8)  | 27.0(-6.0)  | 22.0(-6.2)  |

---

> ### Author Response · Authors · 2022-11-19
> **Response to Reviewer MqVZ (Part 1)**
>
> Thank you for the review.
>
> > Weakness on novelty: "On a technical level, I do not see much new contribution, where most equations are from diffusion model"
> > "The idea to include denoising step with diffusion model to improve classification performance is incremental."
>
> Response: Our similarity in equations only lies in the definition of the diffusion models. On top of the definition, we derived the robust region and robust radius to adversarial attacks, which is not covered in previous diffusion model literature and has the potential to provide a large robust region. To the best of our knowledge, our work is the first to explain why and how the diffusion model purifies adversarial attacks, so the results are all new.
>
> Based on these analyses, we then did numerical experiments to verify our theoretical results. We show that DensePure indeed can enhance certified robustness in practice. To the best of our knowledge, our work is the first one to achieve these, and a large part of our work's novelty is in the theoretical analysis, and the results are all new.
>
> In particular, our contributions are as follows:
> 1. We prove that under constrained data density property (i.e., points with the ground-truth label should have relatively high density and are separated away from points with different labels in the original data distribution, see Theorem 3.3), an adversarial example can be recovered back to the original clean sample with a high probability via the reverse process of a diffusion model.
> 2. In theory, we characterized the robust region for each point by further taking the highest density point in the conditional distribution generated by the reverse process as the reversed sample. We show that the robust region for a given sample under the diffusion model's reverse process is the union of multiple convex sets, each surrounding a region around the ground-truth label. Thus, it has the potential to provide a larger robust region. To the best of our knowledge, this is the first work that characterizes the robust region of using the reverse process of the diffusion model for adversarial purification compared with previous works[1][2], which only provide qualitative explanations. Specifically, we strictly defined the robustness region and radius for the adversarial attack, and we provided closed-form representations of the robustness region and radius. However, previous works [1][2] only provide empirical results or show that the diffusion model will bring an adversarial sample to a neighborhood of a true sample, but the radius of the neighborhood could still be large and even larger than the attack magnitude. Thus their results can not be directly translated into robust regions or radii in our work.
> 3. In practice, we proposed DensePure in practice to verify our theoretical analysis. DensePure indeed verifies our analysis and achieves a state-of-art adversarial purification pipeline directly leveraging the reverse process of a pre-trained diffusion model and majority vote. Within this context, our DensePure framework is not just an experimental trial but also a quantitative proof for our theoretical analysis.
>
>
>
> [1] Nie, Weili, et al. "Diffusion Models for Adversarial Purification." arXiv preprint arXiv:2205.07460 (2022).
>
> [2] Carlini, Nicholas, Florian Tramer, and J. Zico Kolter. "(Certified!!) Adversarial Robustness for Free!." arXiv preprint arXiv:2206.10550 (2022).

---

> ### Author Response · Authors · 2022-11-29
> **Summary of Rebuttal to Reviewer MqVZ**
>
> We really appreciate your valuable comments. We hope that your concerns have been addressed. In particular,
> 1. we would like to highlight that we are the first to provide theoretical analysis to (1) explain why and how the diffusion model purifies adversarial attacks to improve the adversarial robustness and (2) derive the robust region and robust radius of diffusion models, which has the potential to provide a large robust region. Based on these analyses, we then did numerical experiments to verify our theoretical results. We listed our novel contributions in detail.
> 2. we confirmed that we strictly followed all settings in [Carlini 2022] for a fair comparison.
> 3. we added experiments in Appendix D.7 showing that without the diffusion model, the robustness would decrease significantly, which showed the nontriviality of the diffusion model in our framework.
> 4. we think using the pretrained diffusion model and classifier without any finetuning or retraining is indeed excellent. In this way, the users or service providers do not need to retrain the models, and models can provide a "free" lunch for adversarial robustness. Furthermore, our approach can either utilize off-the-shelf models to skip the model training process or train our own models, which gives us more options.
> 5. our theoretical results and the framework can be generalized to any noise distribution other than Gaussian. We choose to use diffusion models based on Gaussian noise to (1) get a more tractable characterization of the robustness of DensePure and (2) make better comparisons with previous works, which are mostly based on Gaussian noise. We leave the extension of our method to non-Gaussian noise-based diffusion models as future work.
>
> As we are inching closer to the final recommendation deadline, we would sincerely appreciate your feedback and score update based on our detailed rebuttal. We have incorporated all the changes in our revised manuscript for your kind consideration. Please also contact us for further clarification.

---

> ### Author Response · Authors · 2022-11-30
> **We will be glad to receive your further feedback**
>
> Dear Reviewer MqVz,
>
> We deeply appreciate your valuable comments.   Since we are approaching the reviewer final recommendation deadline, we would sincerely appreciate your further feedback. We hope that our detailed rebuttal has addressed your concern. If your concern has been addressed, we sincerely appreciate that you can update the score. We have incorporated all the changes in our revised manuscript for your kind consideration. Please feel free to contact us for further clarification.
>
>
> Authors.

---

> ### Author Response · Authors · 2022-12-05
> **We will appreciate it so much for further feedback**
>
> Dear Reviewer MqVz,
>
> As you might know that we are approaching the reviewer final recommendation deadline and it will be very helpful if you can provide us with further feedback on our detailed rebuttal. We deeply appreciate your comments and made detailed responses to all your concerns. We also incorporated all your valuable comments in our revised manuscript. If your concerns have been addressed, we sincerely appreciate that you can update the score. Please feel free to reach out in case you need any clarification.
>
> Best
>
> Authors.

---

### Official Review · Reviewer_ChYc · 2022-10-29

**Confidence:** 4
**Correctness:** 2
**Technical Novelty And Significance:** 2
**Empirical Novelty And Significance:** 2
**Recommendation:** 5

**Clarity, Quality, Novelty And Reproducibility:**

The paper is in general clear and easy to follow. And there seems no reproducibility problem. But the novelty is limited, and the theory cannot support the claim very well.


**Strength And Weaknesses:**

Strength:
- The paper gives some theoretical analysis about the conditional generation, which can help clarify the understanding of the problem.
- The experiments show that the proposed method can improve the robust of the classification of adversarial inputs.

Weakness:
- In general, it seems that the classification of the adversarial samples is overkilled. To improve the robustness, the authors need to run the time-consuming diffusion models more than 10 times to get the conditional generation and then run the classifiers. It is unreasonable.
- The theory part just includes some clarifications of common sense, it hard to get some novel ideas through the definitions or theorems.
    - It is also unclear if it is proper to bound the complex data support  with hyper-balls. For example, in the data manifold, if the data labeled by 1 is supported in a 2-dimensional rectangular like region with large length and width ratio, can we use definition 3.2 or theorem 3.3 to measure the results?
    - In theorem 3.4, KL divergence may not be a good method to measure the similarity of two distributions since it is not a distance.

**Summary Of The Paper:**

This paper proposes to solve the potential classification problem of the adversarial samples with the help of diffusion models. Specifically, the authors firstly give some analysis of the regions of the conditional generation. They claim that the conditionally generated samples will concentrate on the regions around the original adversarial sample. Then through majority voting, the proposed method can help find the correct labels. Experiments show that the proposed method does help improve the performance.

**Summary Of The Review:**

For one thing, the paper help clarify the conditional distribution of the diffusion models, which may help the community; for another, the theoretical contribution seems limited and the proposed method overkills the adversarial classification problem in terms of resource and time.

---

> ### Author Response · Authors · 2022-11-19
> **Response to Reviewer ChYc (Part 3)**
>
> > Clarity, Quality, Novelty And Reproducibility: The paper is in general clear and easy to follow. And there seems no reproducibility problem. But the novelty is limited, and the theory cannot support the claim very well.
> >
> > Summary Of The Review: For one thing, the paper help clarify the conditional distribution of the diffusion models, which may help the community; for another, the theoretical contribution seems limited and the proposed method overkills the adversarial classification problem in terms of resource and time.
>
> Response: Thanks for pointing that out. We want to highlight that we are the first to provide theoretical analysis to (1) explain why and how the diffusion model purifies adversarial attacks to improve the adversarial robustness and (2) derive the robust region and robust radius of diffusion models, which has the potential to provide a large robust region. Based on these analyses, we then did numerical experiments to verify our theoretical results. We show that DensePure indeed can enhance certified robustness in practice. To the best of our knowledge, our work is the first one to achieve these, and a large part of our work's novelty is in the theoretical analysis, and the results are all new.
>
> In particular, our contributions are as follows:
> 1. We prove that under constrained data density property (i.e., points with the ground-truth label should have relatively high density and are separated away from points with different labels in the original data distribution, see Theorem 3.3), an adversarial example can be recovered back to the original clean sample with a high probability via the reverse process of a diffusion model.
> 2. In theory, we characterized the robust region for each point by further taking the highest density point in the conditional distribution generated by the reverse process as the reversed sample. We show that the robust region for a given sample under the diffusion model's reverse process is the union of multiple convex sets, each surrounding a region around the ground-truth label. Thus, it has the potential to provide a larger robust region. To the best of our knowledge, this is the first work that characterizes the robust region of using the reverse process of the diffusion model for adversarial purification compared with previous works[1][2], which only provide qualitative explanations. Specifically, we strictly defined the robustness region and radius for the adversarial attack, and we provided closed-form representations of the robustness region and radius. However, previous works [1][2] only provide empirical results or show that the diffusion model will bring an adversarial sample to a neighborhood of a true sample, but the radius of the neighborhood could still be large and even larger than the attack magnitude. Thus their results can not be directly translated into robust regions or radii in our work.
> 3. In practice, we proposed DensePure in practice to verify our theoretical analysis. DensePure indeed verifies our analysis and achieves a state-of-art adversarial purification pipeline directly leveraging the reverse process of a pre-trained diffusion model and majority vote. Within this context, our DensePure framework is not just an experimental trial but also a quantitative proof for our theoretical analysis.
>
>
> [1] Nie, Weili, et al. "Diffusion Models for Adversarial Purification." arXiv preprint arXiv:2205.07460 (2022).
>
> [2] Carlini, Nicholas, Florian Tramer, and J. Zico Kolter. "(Certified!!) Adversarial Robustness for Free!." arXiv preprint arXiv:2206.10550 (2022).

---

> > ### Comment · Reviewer_ChYc · 2022-12-07
> > **Thanks for the detailed response**
> >
> > - I thank the authors for the detailed responses, especially for the bunch of newly added experiments, it is really a great effort!
> > - But I still cannot get the reason why we need such complex conductions to solve the adversarial robust problem, considering that the diffusion models are pretty huge and complicated. Therefore I decide to stick with my previous ratings.

---

> > > ### Author Response · Authors · 2022-12-07
> > > **Response to the further comments of Reviewer ChYc**
> > >
> > > Thanks for your further comments. We sincerely hope you can consider changing the ratings. The reasons are as follows:
> > > * The adversarial robust problem is not an easy problem. it’s challenging to defend against adversarial perturbations as evidenced by a large amount of literature (thousands of papers) in recent years. Without the diffusion model, on ImageNet, the certified accuracy is still very low in ImageNet.
> > > * DensePure achieves state-of-the-art certified robustness and outperforms existing methods on ImageNet, with a large margin ( 7% improvement on average). Moreover, Compared with non-diffusion model-based methods (e.g., PixelDP, RS, SmoothAdv, MACER, Boosting), it outperforms them by over 12% on average, although those methods introduce additional training time cost. Compared with the off-the-shelf-based method (Denoised, Lee), our method outperforms them by over 37% on average. Those large margins also show the importance of introducing the diffusion model. Given such significant performance improvement, we believe it is worth introducing diffusion models.
> > > * On the other hand, the architecture of Densepure is not complicated. It directly uses the off-the-shelf (pre-trained) diffusion model and classifier. Thus, compared with other baseline methods (e.g., SmoothAdv, etc.), it does not introduce any additional training cost. Thus, it can also directly leverage the existing powerful pre-trained classifiers (e.g., Vision Transformer-based models) without additional training costs, providing a “free lunch” for adversarial robustness.
> > > * Compared with the classifiers, the diffusion model is not super huge. The size of the diffusion model is 60 M for CIFAR-10, which is smaller than the size of the VIT classifier (86M).
> > > * The main goal of our paper is to open a new research direction to understand why and how the diffusion model can improve adversarial robustness. As far as we know, it is one of the first works to show such performance improvement. Although the diffusion model introduces additional time efficiency problems, our newly added experiments in the previous rebuttal have shown great potential to improve time efficiency. Additionally, reducing the time complexity itself is one of the hot topics in the diffusion model literature. There are many studies of fast sampling strategies to reduce the time complexity. With the development of that domain, our method has the potential to be combined with them to further reduce the time efficiency.
> > >
> > > Given the above reasons, we believe it is worth introducing diffusion models for the adversarial robustness community.

---

> > > ### Author Response · Authors · 2022-12-08
> > > **Thanks for further comments.**
> > >
> > > Dear reviewer ChYc,
> > >
> > > We thank for your further comments. We kindly check whether our new response has addressed your concern?  We sincerely hope you can consider changing the ratings.

---

> > > ### Author Response · Authors · 2022-12-12
> > > **Thanks for your comments. Any pending questions?**
> > >
> > > Dear Reviewer,
> > >
> > > The final decision deadline is very soon. We sincerely hope to get your further response. If you do not have any question, we sincerely hope you can consider increasing your ratings.

---

> ### Author Response · Authors · 2022-11-19
> **Response to Reviewer ChYc (Part 2)**
>
> - > The theory part just includes some clarifications of common sense, it hard to get some novel ideas through the definitions or theorems.
>   >
>   > - It is also unclear if it is proper to bound the complex data support with hyper-balls. For example, in the data manifold, if the data labeled by 1 is supported in a 2-dimensional rectangular like region with large length and width ratio, can we use definition 3.2 or theorem 3.3 to measure the results?
>   >
>   > - In theorem 3.4, KL divergence may not be a good method to measure the similarity of two distributions since it is not a distance.
>
> Response: For the first point, yes, we can. In fact, we do not bound the data support and make no assumptions about the data support. Our theoretical results characterize the robust regions with respect to the data support, which are represented as intersections of halfspaces. The intersection of halfspaces can generally represent any convex set [1] (specifically, the robust subregions). Then we use unions of convex robust sub-regions to represent a more general non-convex robust region. In particular, our theoretical results can work not only on rectangular-like regions, but also regions with arbitrary shapes.
>
> For the second point, we can use the total variation $\text{TV}(P, Q)$ as a distance metric for the probability distribution. Here we apply the Pinsker’s inequality $\text{TV}(P, Q) \leq \sqrt{\frac{1}{2} D_{\mathrm{KL}}(P \| Q)}$ to derive an upper bound on the total variation based on KL divergence. We did not include this result in the current version, but we are happy to add it to the final version of the paper. We used KL divergence in the current version because it can be upper bounded by the training loss of the diffusion model. As theorem 3.4 shows, if we can train a good diffusion model with small training loss, we will have a small KL divergence of the two distributions, which implies their similarity, even measured in total variation.
>
> [1] Boyd, Stephen, Stephen P. Boyd, and Lieven Vandenberghe. *Convex optimization*. Cambridge university press, 2004.

---

> ### Author Response · Authors · 2022-11-19
> **Response to Reviewer ChYc (Part 1-2)**
>
> Furthermore, there is a large potential to improve the time efficiency of DensePure. For instance,  we can use K-Consensus Aggregation proposed in the paper [1] to improve the efficiency of the majority voting of DensePure. In this algorithm, if the classification results of the K consecutive reversed samples are the same, an early stop will be triggered.
> We conduct additional experiments by using K-consensus aggregation. We calculate certified robustness for 100 subsamples of CIFAR-10 and ImageNet with 2 sampling steps, a maximum 10 majority votes and consensus threshold k=3. The results are shown as follows:
>
> ​
>
>  CIFAR-10
>
> | Certified Accuracy at $\epsilon$($\%$) | 0.00     | 0.25     | 0.50     | 0.75     | 1.00     | Avg majority votes |
> | -------------------------------------- | -------- | -------- | -------- | -------- | -------- | ------------------ |
> | $\sigma=0.25$                          | 92(+0.0) | 77(+0.0) | 60(+0.0) | 48(-1.0) | 0(+0.0)  | 3.84               |
> | $\sigma=0.50$                          | 74(+0.0) | 65(+0.0) | 53(-1.0) | 45(+0.0) | 40(+0.0) | 4.43               |
> | $\sigma=1.00$                          | 53(+0.0) | 46(+0.0) | 42(+0.0) | 31(+0.0) | 25(+0.0) | 5.49               |
>
>  ImageNet
>
> | Certified Accuracy at $\epsilon$($\%$) | 0.00     | 0.25     | 0.50     | 0.75     | 1.00     | 3.00     | Avg majority votes |
> | -------------------------------------- | -------- | -------- | -------- | -------- | -------- | -------- | ------------------ |
> | $\sigma=0.25$                          | 78(+0.0) | 74(+0.0) | 0(+0.0)  | 0(+0.0)  | 0(+0.0)  | 0(+0.0)  | 3.34               |
> | $\sigma=0.50$                          | 75(+0.0) | 69(+0.0) | 61(+0.0) | 47(+0.0) | 0(+0.0)  | 0(+0.0)  | 3.89               |
> | $\sigma=1.00$                          | 60(+0.0) | 54(+0.0) | 50(+0.0) | 41(+0.0) | 32(+0.0) | 23(+0.0) | 5.23               |
>
>
> The column of avg majority votes means the average of the actual number of majority votes required for our algorithm. For instance,  if the predicted labels of the first 3 reversed samples are the same, the actual majority vote numbers will be 3. The numbers in the bracket are the difference between certified accuracy w/o K-Consensus Aggregation.
>
>
> We observe that comparing with 2 sampling steps 10 majority votes without K-Consensus Aggregation, using K-Consensus Aggregation can make the majority vote process 45%~60% faster with almost no performance degradation with respect to the certified accuracy.
>
>
>
> Moreover, the time efficiency of DensePure can be further improved by accelerating randomized smoothing algorithms or even the sampling speed of diffusion models. For instance:
>
> 1. In paper [1], the adaptive sampling method for randomized smoothing shown in Section 6 can also be used in our DensePure pipeline to make faster-randomized smoothing.
>
> 2. To improve the sampling speed of diffusion models, we can use the distilled diffusion model in paper [2], which can generate high-quality images in only 4 steps.
>
> We will add related discussions in our revision.
>
> [1] Miklos Z Horvath, Mark Niklas Muller, Marc Fischer, and Martin Vechev. Boosting randomized smoothing with variance reduced classifiers. arXiv preprint arXiv:2106.06946, 2021.
>
> [2]Tim Salimans, Jonathan Ho. Progressive Distillation for Fast Sampling of Diffusion Models. ICLR 2022

---

> ### Author Response · Authors · 2022-11-19
> **Response to Reviewer ChYc (Part 1-1)**
>
> Thank you for the constructive feedback.
>
> > **Weakness:**
> >
> > - In general, it seems that the classification of the adversarial samples is overkilled. To improve the robustness, the authors need to run the time-consuming diffusion models more than 10 times to get the conditional generation and then run the classifiers. It is unreasonable.
>
> Response: Thanks for your question. We hope to highlight that the main goal of this paper is to provide a theoretical understanding of why and how the adversarial purification by diffusion models contributes to adversarial robustness, which has not been explored.
> To the best of our knowledge, (1) we are the first to prove that under constrained data density property (i.e., points with the ground-truth label should have relatively high density and are far away from points with different labels), an adversarial example can be recovered back to the original clean sample with a high probability via the reverse process of a diffusion model;
> (2) we are the first to characterize the robust region of diffusion models against adversarial attacks and show that the robust region of a given sample under the diffusion model’s reverse process has the potential to provide a larger robust region.
>
> The certification algorithm, DensePure, mainly aims to empirically verify our theoretical understanding and needs to follow our theoretical understanding. Our empirical results confirm that DensePure can achieve higher certified robustness than existing methods.
>
> To further verify our improvement is non-trivial, we conduct additional experiments by removing the diffusion model from DensePure on ImageNet.  To fairly compare with randomized smoothing without diffusion models under majority votes, we activate droppath in BEiT at the inference stage to support majority votes. The other settings are the same as DensePure. The results are shown in the following table. The number in the bracket is calculated by the robust accuracy of BeiT with majority votes - the robust accuracy of DensePure.
> We show that simply performing majority votes on the BeiT classifier will not result in higher certified robustness.
>
> | Certified Accuracy at $\epsilon$(%) | 0.0         | 0.5         | 1.0        | 1.5        | 2.0        | 3.0        |
> | ----------------------------------- | ----------- | ----------- | ---------- | ---------- | ---------- | ---------- |
> | $\sigma=0.25$                       | 73.8(-10.2) | 58.0(-19.8) | 0.0(+0.0)  | 0.0(+0.0)  | 0.0(+0.0)  | 0.0(+0.0)  |
> | $\sigma=0.50$                       | 9.0(-71.2)  | 7.0(-68.6)  | 4.0(-63.0) | 2.0(-52.6) | 0.0(+0.0)  | 0.0(+0.0)  |
> | $\sigma=1.00$                       | 0.0(-67.8)  | 0.0(-61.4)  | 0.0(-55.6) | 0.0(-50.0) | 0.0(-42.2) | 0.0(-25.8) |
>
>
>
>
> Within this context, how to further improve the time efficiency of DensePure is an open problem. In Figure 4 and Appendix D.5 (Figure C,D,G), we reduce the reverse sampling steps and majority vote numbers to improve efficiency. We show that with even 2 sampling steps and 10 majority votes (which can largely reduce the repeating times), we can still achieve non-trivial certified robustness. Additionally, we also conduct additional experiments with 2 sampling steps and 5 majority votes. The results are shown as follows. We  find that our method still achieves better results than the existing method.
>
> CIFAR-10
>
> | Certified Accuracy at $\epsilon$(%) | 0.00 | 0.25 | 0.50 | 0.75 | 1.00 |
> | ----------------------------------- | ---- | ---- | ---- | ---- | ---- |
> | $\sigma=0.25$                       | 87.6 | 74.8 | 59.2 | 44.6 | 0.0  |
> | $\sigma=0.50$                       | 73.2 | 62.6 | 52.6 | 41.8 | 34.0 |
> | $\sigma=1.00$                       | 53.4 | 44.0 | 35.8 | 30.2 | 24.4 |
>
> ImageNet
>
> | Certified Accuracy at $\epsilon$(%) | 0.0  | 0.5  | 1.0  | 1.5  | 2.0  | 3.0  |
> | ----------------------------------- | ---- | ---- | ---- | ---- | ---- | ---- |
> | $\sigma=0.25$                       | 78   | 74   | 0    | 0    | 0    | 0    |
> | $\sigma=0.50$                       | 75   | 69   | 58   | 47   | 0    | 0    |
> | $\sigma=1.00$                       | 60   | 54   | 49   | 39   | 30   | 22   |

---

> ### Author Response · Authors · 2022-11-29
> **Summary of Rebuttal to Reviewer ChYc**
>
> We really appreciate your valuable comments. We hope that your concerns have been addressed. In particular:
> 1. we would like to highlight that we are the first to provide theoretical analysis to (1) explain why and how the diffusion model purifies adversarial attacks to improve the adversarial robustness and (2) derive the robust region and robust radius of diffusion models, which has the potential to provide a large robust region. Based on these analyses, we then did numerical experiments to verify our theoretical results. We listed our novel contributions in detail.
> 2. we did our experiments for two reasons: (1) to verify our theoretical analysis (2) to show that this new framework outperforms other state-of-the-art adversarial defense methods. We also provided several ways of reducing computational time in the rebuttal, including fewer samplings, K-Consensus method, and accelerating randomized smoothing algorithms.
> 3. to elaborate on our theoretical results, we clarify that our method does not make any assumption on data support. We illustrated how our method can characterize a robust region of any shape by first defining convex subregions and showing that the entire robust region is a union of them. Our work is the first to provide rigorous characterization for such robust regions and radii, and we also show that these regions and radii are large under reasonable assumptions.
>
> As we are inching closer to the final recommendation deadline, we would sincerely appreciate your feedback and score update based on our detailed rebuttal. We have incorporated all the changes in our revised manuscript for your kind consideration. Please also contact us for further clarification.

---

> ### Author Response · Authors · 2022-11-30
> **We will be glad to receive your  further feedback**
>
> Dear Reviewer ChYc,
>
> We deeply appreciate your valuable comments. As you might know that we are approaching the reviewer final recommendation deadline. Thus, we would really appreciate if you can provide us the feedback on our detailed rebuttal. We have incorporated all changes in the revised manuscript for your kind consideration. We have added your valuable comments into our paper to strengthen our manuscript as well. If your concerns have been addressed, we sincerely appreciate that you can update the score. Feel free to reach us out in case you need any clarification.
>
> Best
>
> Authors.

---

> ### Author Response · Authors · 2022-12-05
> **We will appreciate it so much for further feedback**
>
> Dear Reviewer ChYc,
>
> As you might know that we are approaching the reviewer final recommendation deadline and it will be very helpful if you can provide us with further feedback on our detailed rebuttal. We deeply appreciate your comments and made detailed responses to all your concerns. We also incorporated all your valuable comments in our revised manuscript. If your concerns have been addressed, we sincerely appreciate that you can update the score. Please feel free to reach out in case you need any clarification.
>
> Best
>
> Authors.

---

### Author Response · Authors · 2022-11-19
**Revision List of the paper & link to the code**

### Revision List
1. Page 5, Section 3, Remark 3. We added comments illustrating under constrained data density properties (i.e., points with the ground-truth labels should have relatively high density and are separated away from points with different labels), the robust region/ robust radius for points with the ground-truth labels (clean samples) will be large. Since adversarial attack magnitude is typically small, the adversarial sample will be in the robust region/ robust radius with high probability and thus can be recovered to the clean sample with high probability.
2. Page 5, Section 3, Remark 4. We added an upper bound of the total variation distance between the true conditional distribution and the conditional distribution recovered by the diffusion model using the training loss. This is an extension of the results in Thm 3.4.
3. Page 8, Section 5.1. We added ablation study experiment results in Appendix D.7 to show that removing the diffusion model from DensPure significantly deteriorates the performance and thus the diffusion model is necessary for DensePure.
4. Page 8, Section 5.2. We added experiment results in Appendix D.8 using the K consensus technique to approximate the majority vote, which accelerates the voting process by 45% ~ 60% with a negligible performance drop.
5. Appendix D.7 shows the necessity of the diffusion model in DensePure; Appendix D.8 shows that K-consensus aggregation can significantly improve the voting efficiency with merely a negligible performance drop; Appendix D.9 shows that reducing the sampling step and majority votes to 2 and 5 can improve the voting efficiency significantly while still outperform existing methods.
7. In Appendix D.10, We increased the ImageNet test sampling number to 500 and updated the experiment results.

### DensePure Code
https://anonymous.4open.science/r/DensePure-4D50/README.md

---

### Decision · Program_Chairs · 2023-01-20

**Decision:**

Accept: poster

**Justification For Why Not Higher Score:**

The technical contribution seems limited and the proposed method overkills the adversarial classification problem in terms of resource and time.

**Justification For Why Not Lower Score:**

The theoretically analysis of why diffusion model is able to perform purification task is well grounded and important.

**Metareview: Summary, Strengths And Weaknesses:**

Summary:  This paper studies the conditions under which diffusion model can work well for purification of adversarially perturbed samples. Then a simple diffusion-based purification method named DensePure is proposed by using majority vote. The extensive experiments demonstrate the effectiveness of DensePure by evaluating its certified robustness given a standard model and show it is consistently better than existing methods on ImageNet.

Strengths: Theoretical strength: This paper theoretically analyzes why and how diffusion-based purification model can enhance the adversarial robustness of a given classifier for the first time. The robust region of a sample is characterized as the union of several convex sub-robust regions. The proposed DensePure performs consistently better than existing methods on ImageNet, with 7% improvement on average.

Weakness: DensePure is really time-consuming and this drawback may prevent it from further applications to large datasets. Both diffusion model for purification and random smoothing technique are existing methods, and thus DensePure is a straightforward combination of these two methods.

**Note From Pc:**

if the above contains the word "oral" or "spotlight" please see: "oral" presentation means -> notable-top-5% and "spotlight" means -> notable-top-25%. As stated in our emails, we are disassociating presentation type from AC recommendations